# The Current Status and Future Prospects of KAGRA, the Large-Scale Cryogenic Gravitational Wave Telescope Built in the Kamioka Underground

Homare Abe [1], Tomotada Akutsu [2], Masaki Ando [3], Akito Araya [4], Naoki Aritomi [3], Hideki Asada [5], Yoichi Aso [6], Sangwook Bae [7], Rishabh Bajpai [8], Kipp Cannon [9], Zhoujian Cao [10], Eleonora Capocasa [2], Man Leong Chan [11], Dan Chen [6], Yi-Ru Chen [12], Marc Eisenmann [2], Raffaele Flaminio [13], Heather K. Fong [9], Yuta Fujikawa [14], Yuya Fujimoto [15], I. Putu Wira Hadiputrawan [16], Sadakazu Haino [17], Wenbiao Han [18], Kazuhiro Hayama [11], Yoshiaki Himemoto [19], Naoatsu Hirata [2], Chiaki Hirose [14], Tsung-Chieh Ho [16], Bin-Hua Hsieh [20], He-Feng Hsieh [21], Chia-Hsuan Hsiung [22], Hsiang-Yu Huang [17], Panwei Huang [23], Yao-Chin Huang [12], Yun-Jing Huang [17], David C. Y. Hui [24], Kohei Inayoshi [25], Yuki Inoue [16], Yousuke Itoh [15], Pil-Jong Jung [26], Takaaki Kajita [20], Masahiro Kamiizumi [20], Nobuyuki Kanda [15], Takashi Kato [20], Chunglee Kim [27], Jaewan Kim [28], Young-Min Kim [29], Yuichiro Kobayashi [15], Kazunori Kohri [30], Keiko Kokeyama [31], Albert K. H. Kong [21], Naoki Koyama [14], Chihiro Kozakai [6], Jun'ya Kume [9], Sachiko Kuroyanagi [32], Kyujin Kwak [29], Eunsub Lee [20], Hyung Won Lee [33], Ray-Kuang Lee [12], Matteo Leonardi [2], Kwan-Lok Li [34], Pengbo Li [35], Lupin Chun-Che Lin [29], Chun-Yu Lin [36], En-Tzu Lin [21], Hong-Lin Lin [16], Guo-Chin Liu [22], Ling-Wei Luo [17], Miftahul Ma'arif [16], Yuta Michimura [3], Norikatsu Mio [37], Osamu Miyakawa [20], Kouseki Miyo [20], Shinji Miyoki [20], Nozomi Morisue [15], Kouji Nakamura [2], Hiroyuki Nakano [38], Masayuki Nakano [39,*,†] 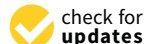, Tatsuya Narikawa [20], Lan Nguyen Quynh [40], Takumi Nishimoto [20], Atsushi Nishizawa [9], Yoshihisa Obayashi [20], Kwangmin Oh [24], Masatake Ohashi [20], Tomoya Ohashi [15], Masashi Ohkawa [14], Yoshihiro Okutani [41], Ken-ichi Oohara [20], Shoichi Oshino [20], Kuo-Chuan Pan [12], Alessandro Parisi [22], June Gyu Park [42], Fabián E. Peña Arellano [20], Surojit Saha [21], Kazuki Sakai [43], Takahiro Sawada [15], Yuichiro Sekiguchi [44], Lijing Shao [25], Yutaka Shikano [45], Hirotaka Shimizu [46], Katsuhiko Shimode [20], Hisaaki Shinkai [47], Ayaka Shoda [2], Kentaro Somiya [1], Inhyeok Song [21], Ryosuke Sugimoto [48], Jishnu Suresh [20], Takamasa Suzuki [1], Takanori Suzuki [14], Toshikazu Suzuki [20], Hideyuki Tagoshi [20], Hirotaka Takahashi [49], Ryutaro Takahashi [2], Hiroki Takeda [3], Mei Takeda [15], Atsushi Taruya [50], Takayuki Tomaru [2], Tomonobu Tomura [20], Lucia Trozzo [20], Terrence T. L. Tsang [51], Satoshi Tsuchida [15], Takuya Tsutsui [9], Darkhan Tuyenbayev [15], Nami Uchikata [20], Takashi Uchiyama [20], Tomoyuki Uehara [52], Koh Ueno [9], Takafumi Ushiba [20,*,†] , Maurice H. P. M. van Putten [53], Tatsuki Washimi [6,*,†], Chien-Ming Wu [12], Hsun-Chung Wu [12], Tomohiro Yamada [46], Kazuhiro Yamamoto [54], Takahiro Yamamoto [20], Ryo Yamazaki [41], Shu-Wei Yeh [12], Jun'ichi Yokoyama [9], Takaaki Yokozawa [20], Hirotaka Yuzurihara [20], Simon Zeidler [55] and Yuhang Zhao [20]

1   Graduate School of Science, Tokyo Institute of Technology, Meguro-ku, Tokyo 152-8551, Japan
2   Gravitational Wave Science Project, National Astronomical Observatory of Japan (NAOJ), Tokyo 181-8588, Japan
3   Department of Physics, The University of Tokyo, Bunkyo-ku, Tokyo 113-0033, Japan
4   Earthquake Research Institute, The University of Tokyo, Bunkyo-ku, Tokyo 113-0032, Japan
5   Department of Mathematics and Physics, Gravitational Wave Science Project, Hirosaki University, Hirosaki 036-8561, Japan
6   Kamioka Branch, National Astronomical Observatory of Japan (NAOJ), Kamioka-cho, Gifu 506-1205, Japan
7   Korea Institute of Science and Technology Information (KISTI), Yuseong-gu, Daejeon 34141, Korea
8   School of High Energy Accelerator Science, The Graduate University for Advanced Studies (SOKENDAI), Ibaraki, Tsukuba 305-0801, Japan

9    Research Center for the Early Universe (RESCEU), The University of Tokyo,
     Bunkyo-ku, Tokyo 113-0033, Japan
10   Department of Astronomy, Beijing Normal University, Beijing 100875, China
11   Department of Applied Physics, Fukuoka University, Jonan, Fukuoka 814-0180, Japan
12   Department of Physics, National Tsing Hua University, Hsinchu 30013, Taiwan
13   Laboratoire d'Annecy de Physique des Particules (LAPP), University Grenoble Alpes,
     Université Savoie Mont Blanc, CNRS/IN2P3, F-74941 Annecy, France
14   Faculty of Engineering, Niigata University, Nishi-ku, Niigata 950-2181, Japan
15   Department of Physics, Graduate School of Science, Osaka City University, Sumiyoshi-ku,
     Osaka 558-8585, Japan
16   Center for High Energy and High Field Physics, Department of Physics, National Central University,
     Taoyuan City 32001, Taiwan
17   Institute of Physics, Academia Sinica, Taipei 11529, Taiwan
18   Shanghai Astronomical Observatory, Chinese Academy of Sciences, Shanghai 200030, China
19   College of Industrial Technology, Nihon University, Narashino 275-8575, Japan
20   Institute for Cosmic Ray Research (ICRR), KAGRA Observatory, The University of Tokyo,
     Kashiwa 277-8582, Japan
21   Institute of Astronomy, National Tsing Hua University, Hsinchu 30013, Taiwan
22   Department of Physics, Tamkang University, New Taipei City 25137, Taiwan
23   State Key Laboratory of Magnetic Resonance and Atomic and Molecular Physics, Innovation Academy for
     Precision Measurement Science and Technology (APM), Chinese Academy of Sciences, Wuhan 430071, China
24   Department of Astronomy & Space Science, Chungnam National University,
     Yuseong-gu, Daejeon 34134, Korea
25   Kavli Institute for Astronomy and Astrophysics, Peking University, Haidian District, Beijing 100871, China
26   National Institute for Mathematical Sciences, Yuseong-gu, Daejeon 34047, Korea
27   Department of Physics, Ewha Womans University, Seodaemun-gu, Seoul 03760, Korea
28   Department of Physics, Myongji University, Yongin 17058, Korea
29   Department of Physics, Ulsan National Institute of Science and Technology (UNIST),
     Ulju-gun, Ulsan 44919, Korea
30   Institute of Particle and Nuclear Studies (IPNS), High Energy Accelerator Research Organization (KEK),
     Ibaraki, Tsukuba 305-0801, Japan
31   School of Physics and Astronomy, Cardiff University, Cardiff CF24 3AA, UK
32   Instituto de Fisica Teorica, 28049 Madrid, Spain
33   Department of Computer Simulation, Inje University, Gyeongsangnam-do, Gimhae 50834, Korea
34   Department of Physics, National Cheng Kung University, Tainan City 70101, Taiwan
35   School of Physics and Technology, Wuhan University, Wuhan 430072, China
36   National Center for High-Performance Computing, National Applied Research Laboratories, Hsinchu Science
     Park, Hsinchu 30076, Taiwan
37   Institute for Photon Science and Technology, The University of Tokyo, Bunkyo-ku, Tokyo 113-8656, Japan
38   Faculty of Law, Ryukoku University, Fushimi-ku, Kyoto 612-8577, Japan
39   California Institute of Technology, Pasadena, CA 91125, USA
40   Department of Physics, University of Notre Dame, Notre Dame, IN 46556, USA
41   Department of Physics and Mathematics, Aoyama Gakuin University, Kanagawa, Sagamihara 252-5258, Japan
42   Korea Astronomy and Space Science Institute (KASI), Yuseong-gu, Daejeon 34055, Korea
43   Department of Electronic Control Engineering, National Institute of Technology, Nagaoka College,
     Niigata 940-8532, Japan
44   Faculty of Science, Toho University, Funabashi 274-8510, Japan
45   Graduate School of Science and Technology, Gunma University, Maebashi 371-8510, Japan
46   Accelerator Laboratory, High Energy Accelerator Research Organization (KEK),
     Ibaraki, Tsukuba 305-0801, Japan
47   Faculty of Information Science and Technology, Osaka Institute of Technology, Hirakata 573-0196, Japan
48   Department of Space and Astronautical Science, The Graduate University for Advanced Studies
     (SOKENDAI), Kanagawa, Sagamihara 252-5210, Japan
49   Research Center for Space Science, Advanced Research Laboratories, Tokyo City University,
     Setagaya, Tokyo 158-0082, Japan
50   Yukawa Institute for Theoretical Physics (YITP), Kyoto University, Sakyou-ku, Kyoto 606-8502, Japan
51   Faculty of Science, Department of Physics, The Chinese University of Hong Kong, Hong Kong 518172, China
52   Department of Communications Engineering, National Defense Academy of Japan,
     Kanagawa, Yokosuka 239-8686, Japan
53   Department of Physics and Astronomy, Sejong University, Gwangjin-gu, Seoul 143-747, Korea
54   Faculty of Science, University of Toyama, Toyama 930-8555, Japan
55   Department of Physics, Rikkyo University, Toshima-ku, Tokyo 171-8501, Japan

\*    Correspondence: masayuki@caltech.edu (M.N.); ushiba@icrr.u-tokyo.ac.jp (T.U.);
tatsuki.washimi@nao.ac.jp (T.W.)

†    These authors contributed equally to this work.

**Abstract:** KAGRA is a gravitational-wave (GW) detector constructed in Japan with two unique key features: It was constructed underground, and the test-mass mirrors are cooled to cryogenic temperatures. These features are not included in other kilometer-scale detectors but will be adopted in future detectors such as the Einstein Telescope. KAGRA performed its first joint observation run with GEO600 in 2020. In this observation, the sensitivity of KAGRA to GWs was inferior to that of other kilometer-scale detectors such as LIGO and Virgo. However, further upgrades to the detector are ongoing to reach the sensitivity for detecting GWs in the next observation run, which is scheduled for 2022. In this article, the current situation, sensitivity, and future perspectives are reviewed.

**Keywords:** gravitational wave detector; laser interferometer; cryogenics; underground

## 1. Introduction

A gravitational wave (GW) is a physical phenomenon predicted by Einstein in his general theory of relativity in 1916. A GW is a wave of spacetime distortion caused by the motion of mass. It travels at the speed of light. The amplitude and waveform of a GW depend on the acceleration and mass of the source. A heavier mass that changes its motion at a faster rate generates stronger GWs. For instance, astronomically massive phenomena, such as a merger of binary neutron stars or black holes, are powerful sources and represent the main targets of GW detectors that are currently in operation. Because GWs have a different emission process than other measures used in astronomy, such as visible light, X-rays, infrared rays, radio waves, cosmic rays, and neutrinos, unique information can be obtained by observing GWs. In addition, because the interaction between GWs and objects is relatively weak, GWs can propagate through space without being scattered or absorbed by objects. Therefore, even GWs generated just after the birth of the universe can reach Earth, and they are therefore expected to act as probes for the history of the universe using future space GW detectors.

GWs distort spacetime and result in a fluctuation of the distance between two points; therefore, the detection of GWs is possible by precisely measuring this distance fluctuation. However, as mentioned above, the amplitude of the distance variation can be as small as $10^{-18}$ m for current GW detectors, which makes GW detection challenging. Thus far, an optical interferometer is the only instrument that allows for direct GW detection. An interferometer is an L-shaped optical instrument that can convert the arm length fluctuation caused by GWs into a laser intensity fluctuation. A GW interaction can also be described as the tidal force, and the L-shaped interferometer can detect the tidal force efficiently as a differential component of the arm length fluctuation. Because the distance fluctuation is proportional to the distance between two points, a longer-arm interferometer is more sensitive to GW interactions. Thus, GW detectors have continued to increase in size over the last few decades, and current GW detectors have a large kilometer-scale arm. Nevertheless, typical GWs from a target source cause arm-length fluctuations of only $10^{-18}$ m. Currently, the GW detection network consists of two Advanced LIGOs in the US [1], an Advanced Virgo in Italy [2], and the GEO600 in Germany [3].

The Advanced LIGO and Advanced Virgo detectors are interferometric GW detectors with a kilometer-scale arm length, whereas GEO600 has a 600-m arm length. The Advanced LIGO detector succeeded in the world's first detection of GWs in 2015, opening up a new astronomical field of GW astronomy. The first GW event observed by the two Advanced LIGO detectors, GW150914 [4], was the merger of binary black holes. Since then, the GW observation network with Virgo has observed several dozen GW events [5,6], including GW170817 [7], a GW emitted by the merger of binary neutron stars.

After the development of three kilometer-scale detectors, the construction of a kilometer-scale GW observatory began in Japan. This detector, named KAGRA, has been under construction since 2012, and it achieved its first joint observation run with other observatories in April 2020 [8]. KAGRA is currently being upgraded for further sensitivity improvements. KAGRA is built underground and uses cryogenic mirrors to lower its thermal noise to improve sensitivity. These features differ from those of other GW detectors. While the Advanced LIGO and Virgo are considered second-generation detectors, these features make KAGRA a 2.5 generation detector, that is, the intermediate generation before third-generation detectors, such as the Einstein Telescope (ET) [9] and Cosmic Explorer (CE) [10]. In this article, the design concept of KAGRA, its performance during the first joint observation run, an evaluation of new technologies and the underground environment, and future plans are described.

## 2. The Design of KAGRA

### 2.1. Location

The KAGRA experimental site is located under Mount Ikenoyama (elevation of 1369 m), Gifu Prefecture, Japan. Figure 1 presents a schematic of the KAGRA experimental site. It consists of three stations (Corner, X-end, and Y-end), two arm tunnels (X-arm and Y-arm) 3 km in length, and several access tunnels. Under the same mountain, there are many neutrino experiments (Super-Kamiokande [11], KamLAND [12], and CANDLES [13]), dark matter experiments (NEWAGE [14], PICOLON [15], and XMASS (closed) [16]), and other R&D experiments, including those of CLIO [17], which is the prototype of KAGRA.

Japan is famous for its frequent earthquakes, especially on the side facing the Pacific Ocean, where two tectonic plates converge. However, earthquake waves are weakened when they propagate across the Tateyama mountain range, standing northeast of the KAGRA site. This is because the low-density ground (1.4–2.2 g/cm$^3$; for example, the normal area is 2.6 g/cm$^3$) is distributed at an altitude of approximately $-5$ km acts as a cushion [18,19], minimizing the site's propensity for earthquakes.

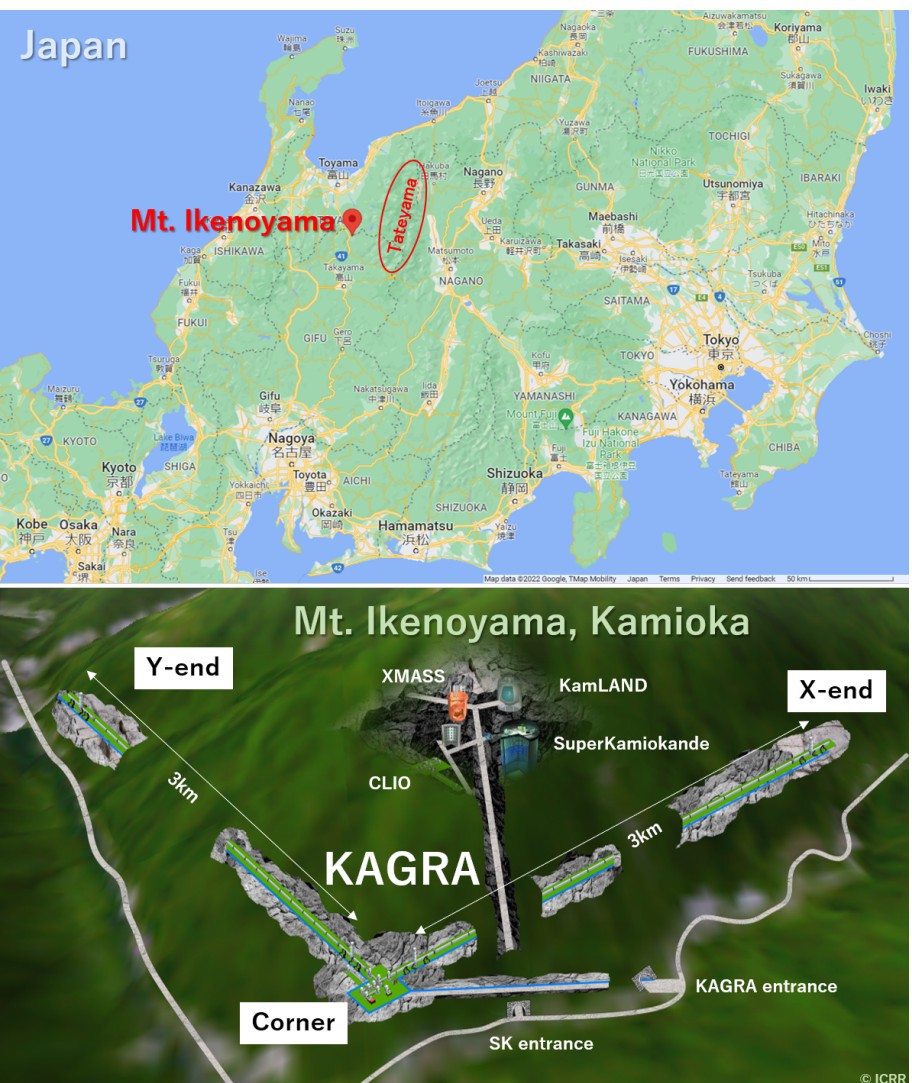

**Figure 1.** Location and schematic view of the KAGRA experimental site [20].

*2.2. Interferometer Configuration*

The simplest configuration of an interferometer is called a Michelson interferometer (MICH, shown in Figure 2a). A Michelson interferometer consists of a laser source, beam splitter (BS) that splits the light into two paths, end mirrors (or end test masses—ETMs) to reflect the split light back to the BS, and photodetector (PD) for measuring the intensity of the recombined light on the BS. The differential component of the variation in the distance from the BS to each ETM (called the arm length) changes the relative phase of the reflected light beams and, consequently, the intensity of the recombined light fluctuates.

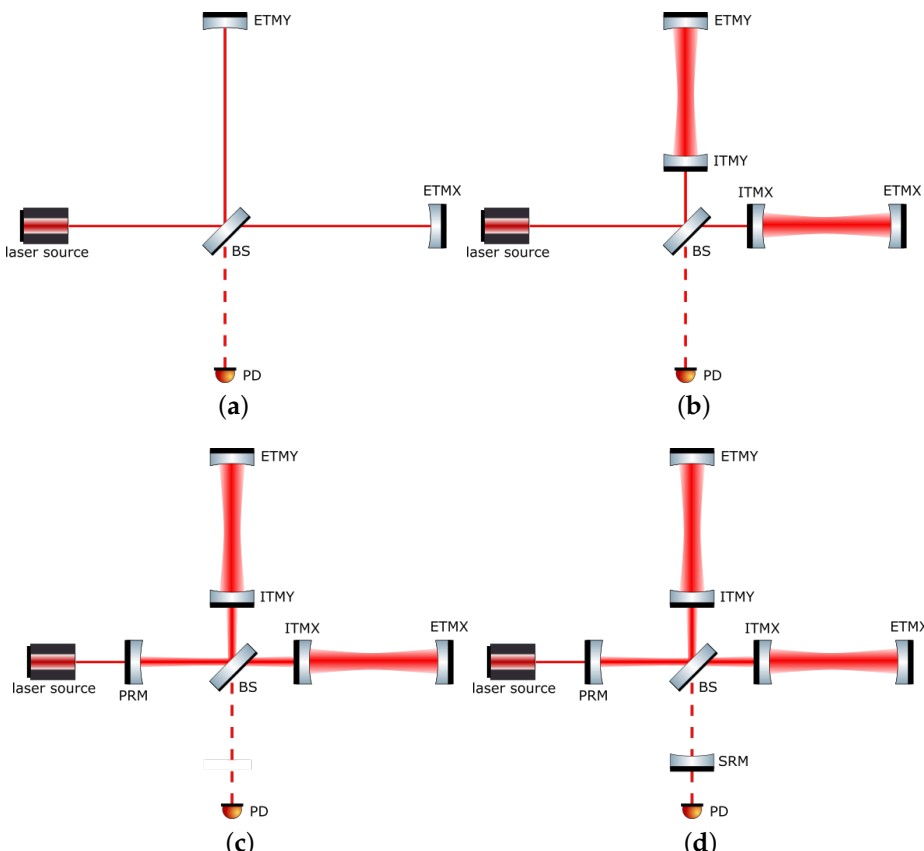

**Figure 2.** Schematic image of the interferometer configuration. A basic Michelson interferometer (**a**) consists of a laser source, beam splitter (BS), end test masses (ETMX and ETMY), and photodetector (PD). The Fabry–Pérot arm cavities, which are composed of ETMs and input test masses (ITMX and ITMY), extend the effective arm length (**b**). The power-recycling mirror (PRM) and signal-recycling mirror (SRM) improve the sensitivity using the power-recycling technique and resonant sideband extraction technique, respectively (**c**,**d**). (**a**) Michelson interferometer (MICH). (**b**) Fabry–Pérot Michelson interferometer (FPMI). (**c**) Power-recycling FPMI (PRFPMI). (**d**) Resonant sideband extraction (RSE) interferometer.

Because interferometers with a longer arm are more sensitive to GWs, KAGRA has 3 km-long arms, as in the other GW observatories. However, because a Michelson interferometer with an arm length of the kilometer order is not sufficient to achieve the required sensitivity for detecting a GW, current GW observatories combine multiple optical cavities with a Michelson interferometer to improve sensitivity, as shown in Figure 2.

First, two Fabry–Pérot optical cavities are incorporated into the arms, forming a configuration called the Fabry–Pérot Michelson interferometer (FPMI, shown in Figure 2b). These arm cavities are composed of input mirrors (or input test masses—ITMs) and ETMs, and they extend the effective arm length of the interferometer. In the case of KAGRA, the arm cavities extend the effective arm length by a factor of approximately 1000, resulting in a 1000-fold increase in the GW sensitivity.

Second, another optical cavity, called the power-recycling cavity, is composed of a power-recycling mirror (PRM) and ITMs. In this power-recycling FPMI configuration (PRFPMI, shown in Figure 2c), the PRM reflects the returning laser beam from the FPMI, and the internal power of the interferometer is amplified [21]. Because the signal-to-noise ratio (SNR) of the GW signal to the quantum shot noise is proportional to the inverse of the square root of the laser power circulating in the arm cavities, the power-recycling cavity further improves sensitivity. In the case of KAGRA, the power-recycling cavity amplifies the laser power in the interferometer by a factor of ten [22].

Then, the other optical cavity is added to the signal port, and it configures the resonant sideband extraction (RSE) interferometer [23] (shown in Figure 2d). The additional optical cavity, called a signal-recycling cavity, is composed of the PRFPMI and a signal-recycling mirror (SRM) placed between the BS and PD. The RSE interferometer works to compensate for the signal cancellation at high frequencies, which is a drawback of the PRFPMI, using the resonant sideband extraction technique. For GWs with a frequency higher than the storage time of laser light in the arm cavity, the obtained signal is averaged, resulting in a weak signal and low sensitivity. The signal-recycling cavity in the RSE interferometer allows for the extraction of the signal sideband of the GWs before cancellation. Furthermore, by detuning the length of the signal-recycling cavity microscopically from the resonance point, the signal-recycling cavity can significantly improve the sensitivity in a specific frequency band. This technique is called detuned RSE [24].

For GW detection, all of the optical resonators must be held in resonance. To achieve this, the distance between the mirrors of the interferometer must be precisely controlled with an accuracy of a few hundred picometers. GW detection is possible only when the interferometer is maintained in a resonant state, which is referred to as "the interferometer is locked".

During the last observation run, the interferometer configuration of KAGRA was that of the PRFPMI. For future observations, a signal-recycling cavity will be installed, and KAGRA will be operated as an RSE interferometer. The design sensitivity was also calculated with the RSE interferometer configuration. The successful operation of the RSE interferometer is one of the major milestones for KAGRA.

### 2.3. Design Sensitivity

GWs are detected as arm length fluctuations; however, their magnitude is very small, that is, the fluctuations are only of the order of $10^{-18}$ m. Thus, even a tiny noise can mask the GW signal. The process for improving the sensitivity of GW detectors can be summarized as noise reduction. The sensitivity curves of the KAGRA design are shown in Figure 3 [25], which presents the fundamental noises that limit the sensitivity of KAGRA. Noise in GW detectors can be roughly divided into two categories based on its effects on the detectors. One is noise that physically shakes the arm's length, which is indistinguishable from arm length fluctuations due to GWs. As shown in Figure 3, seismic, thermal, and quantum radiation pressure noises are categorized as this type of noise. The others, which are represented by quantum shot noise in the sensitivity curve, are those that do not actually shake the test masses but produce a signal resembling an arm length fluctuation, which again makes it indistinguishable from the actual arm length fluctuation. Because these noises have different frequency characteristics, the design sensitivity is limited by the different noises in each frequency band. Roughly speaking, the former noise limits the sensitivity at lower frequencies, since it needs to shake things up, and it gets smaller at higher frequencies, whereas the latter noise is dominant at higher frequencies. Although fundamental noise appears even in a physically ideal detector, a GW detector's sensitivity can easily be polluted by noise from non-ideal features of the system. This type of noise is called technical noise. The technical noise will eventually be reduced to a point lower than the fundamental noise and will not limit sensitivity.

Seismic noise limits the design sensitivity in the frequency band below 5 Hz. The Earth's surface vibrates by approximately 1 μm even in the absence of an earthquake, and the mirror of an interferometer placed on the ground is not immune. In addition, it is difficult to reduce seismic vibration itself. Therefore, in a GW detector, the mirrors are suspended by a pendulum to reduce the transmission of seismic vibrations to the mirrors because seismic noise is attenuated proportional to $f^{-2N}$ above the resonant frequencies of the pendulum, where $N$ is the number of pendulum stages. In the case of KAGRA, the effect of seismic vibration is also minimized by its underground construction, as explained in the next section. The seismic noise is estimated by considering the seismic vibrations

of the ground in the mine, the seismic isolation performance of the pendulums, and the coupling of 1/200 from the vertical seismic vibration, as shown in Figure 3.

Thermal noise, which is caused by the thermal vibration of molecules, is a type of noise that limits the sensitivity at low frequencies below 140 Hz. The thermal vibration of molecules causes the motion of mirrors, their surfaces, and their suspensions, thereby inhibiting GW detection. The thermal noise of a mirror is proportional to the inverse of the square root of the frequencies and limits the sensitivity between 50 and 140 Hz. The thermal noise of a mirror suspension has a noise floor proportional to the inverse of the square of the frequencies and limits the sensitivity below 50 Hz. Several peaks in the thermal noise of the suspension are due to the mechanical resonance of the suspension. The details are summarized in [8]. There are several ways to reduce thermal noise. One is to increase the beam spot size on the mirror to reduce the mirror's thermal noise, which is caused by the vibration of the mirror substrates and coatings, because the correlation of the thermal mirror surfaces between two distant points is small. Another is to use low-mechanical-loss materials for mirrors and their suspensions because thermal noise is proportional to the square root of the mechanical losses of the system [26]. A further method is to reduce the vibration of the molecules by cooling the test-mass mirrors and their suspensions to cryogenic temperatures, as described in the following section. Because this is the first kilometer-scale cryogenic interferometer, the demonstration of cryogenic technology is highly anticipated and is expected to be introduced in future GW detectors.

In the frequency band between 5 and 80 Hz, the sensitivity is limited by quantum radiation pressure noise, which is included as quantum noise in Figure 3. The reflection of light is described in quantum mechanics as the collision of photons on a mirror's surface, which exerts a force called radiation pressure. Because the number of photons in light exhibits quantum fluctuations, the radiation pressure also exhibits inevitable fluctuations, and these fluctuations cause quantum radiation pressure noise.

Quantum shot noise limits the design sensitivity at frequencies above 100 Hz, which is another noise included as quantum noise in Figure 3. A photodetector detects the light intensity by counting the number of photons, which causes its output to fluctuate owing to uncertainty in the number of photons. As described above, an interferometer is an instrument that converts arm-length fluctuations into laser-intensity fluctuations. Therefore, quantum shot noise is inevitable and difficult to reduce in GW detectors.

Both shot and radiation pressure noises are caused by quantum fluctuations of light; however, the relationship between the laser power and SNR is the opposite: the SNR of the radiation pressure noise is proportional to the square root of the power, whereas the SNR of the shot noise is inversely proportional. The frequency response is also different in the two noises: the radiation pressure noise is inversely proportional to the square of the frequency, whereas the shot noise has no frequency dependence, except for its deterioration at high frequencies owing to low-pass filtering for the signal of the optical cavity. Therefore, the optimal laser power is determined by the targeted frequency band. To increase the sensitivity in the high-frequency band, it is necessary to increase the laser power to reduce the shot noise. In the case of the current GW detectors, whose target frequencies are approximately 100 Hz, the intracavity power is designed to be of the order of 1 MW.

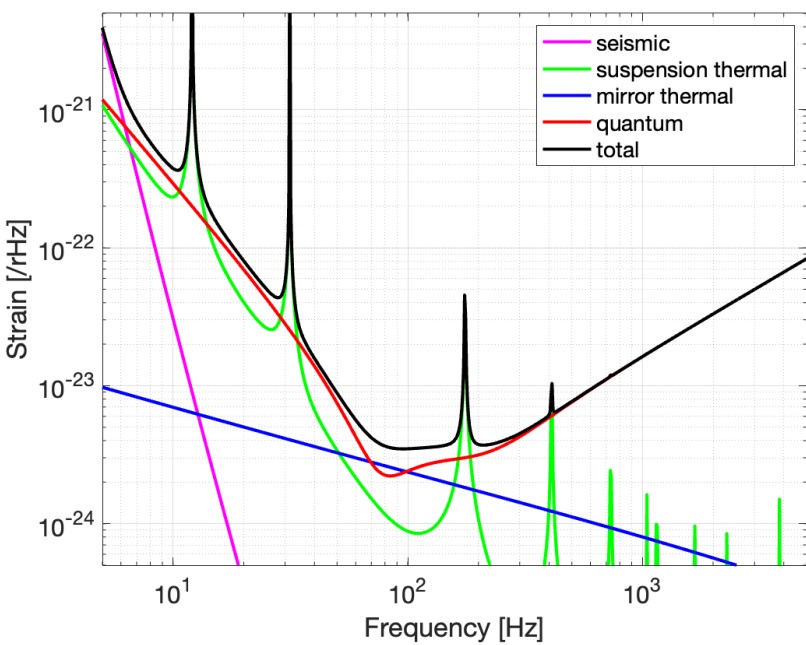

**Figure 3.** Design sensitivity curve of KAGRA [25]. The horizontal axis shows the frequency, and the vertical axis shows the detectable GW signal amplitude. Several fundamental noises are shown: seismic noise, mirror and suspension thermal noise, and quantum noise. Quantum noise is shown as the sum of the quantum radiation pressure noise and shot noise.

Technical noises are caused by imperfections in the interferometer. For example, in an ideal detector, the wavelength of the laser light is considered to contain no fluctuations, other than quantum fluctuations. However, actual laser light exhibits classical wavelength fluctuations, which cause sensing noise. In many cases, the amplitude of these noises, and sometimes even their presence, is difficult to predict. While the magnitude of the fundamental noise is determined by the interferometer design, the magnitude of technical noise is determined by the performance of the interferometer as a whole system; therefore, the interferometer's performance must be optimized after being operated to improve the sensitivity. This sensitivity improvement process is called noise hunting and is the main task in the sensitivity improvement of GW detectors.

### 2.4. Key Features of KAGRA

The fundamental noises described above were reduced by designing KAGRA with two key features. One was the utilization of an underground site to reduce seismic noise. The other was the utilization of cryogenic mirrors to reduce thermal noise. The details are summarized in the following subsections.

### 2.4.1. Underground

Seismic noise is problematic for ground-based GW detectors for two main reasons. The first is sensitivity degradation, as mentioned in the previous section. This can be caused not only by longitudinal motion but also by angular motions of the mirrors through angular-to-longitudinal coupling. The other is the duty-cycle deterioration caused by the angular motions of the mirrors, which disturbs the locking of the interferometer. Owing to the 3 km-long arm, which amplifies the tiny angular motions of the mirrors with respect to the motion of the beam spot, stable operation of the GW detector becomes difficult. There are several technical approaches to reducing these effects, such as utilizing passive vibration isolation with multi-stage suspension and active vibration isolation with inertial sensors; however, further reduction of seismic noise is important for current and future

GW detectors. Because underground sites have low levels of seismic motion with respect to the ground surface, building GW detectors at underground sites is highly beneficial.

The site search for KAGRA was conducted in the late 1990s, and Kamioka in Gifu Prefecture, Japan, was finally selected [27], which is the same location as the two prototype interferometers, a 20 m-scale interferometer, LISM [28], and a 100-m cryogenic interferometer, CLIO [17]. KAGRA was constructed in a horizontal tunnel excavated from a mountain and consists of two 3 km arm tunnels with 1/300 slopes, two end stations, and one corner station. All the stations are located more than 200 m below the surface of the mountain, where the seismic motion is significantly low based on past experience with CLIO [27]. After construction, the seismic motion of the KAGRA site was measured, as shown in Figure 4. Because the seismic motion of the KAGRA site at the observation bands was significantly smaller than that of TAMA300 [29] around the suburbs of Tokyo, the KAGRA location is highly advantageous in terms of seismic noise.

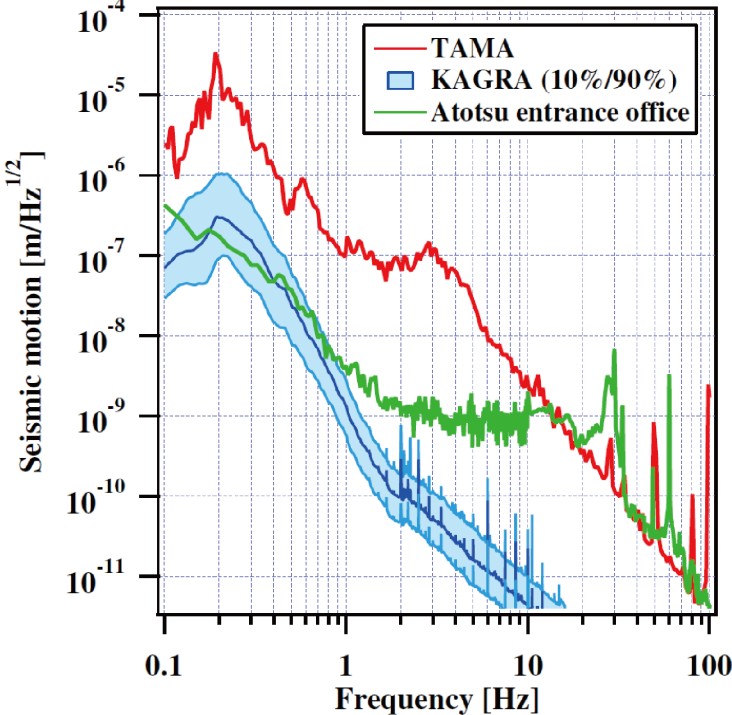

**Figure 4.** Seismic noise at the TAMA300 site (TAMA), the entrance of the KAGRA tunnel (Atotsu entrance), and the KAGRA site (KAGRA). For the seismic noise of the KAGRA site, the 10th and 90th percentile lines (upper and lower cyan lines, respectively) are shown in the figure. Reprinted with permission from Ref. [8], ©2021 Oxford University Press.

Another important benefit of lower seismic motion is the stability of the interferometer lock, because large mirror motions, especially those below 10 Hz, often cause lock loss in the interferometer. In addition, large motions make it difficult to lock the interferometer. Because seismic motion below 10 Hz at the underground site is smaller than that on the ground surface, the KAGRA site has an advantage in terms of stability and sensitivity. In particular, ground motion of approximately 0.2 Hz, which is called microseismic noise, is problematic; however, microseismic noise at the KAGRA site is approximately ten times smaller than that at the TAMA300 site, even in the 90th percentile, as shown in Figure 4. Therefore, the KAGRA site also has the benefit of seismic motion in microseismic bands between 0.1 and 0.4 Hz.

An additional advantage of underground sites is the ability to construct a huge suspension without tall support structures. Because a longer suspension has a better vibration isolation performance in the observation band, it is beneficial to utilize tall suspensions to reduce seismic noise. To achieve a long suspension on the ground, it is

necessary to construct a tall support structure and hang the mirrors from the top, as in the superattenuator in Virgo [30]. In contrast, it is unnecessary to build a tall support in KAGRA because KAGRA involves excavation of a two-story tunnel, and the mirrors can be hung from the second floor. Therefore, although the total height of the test-mass suspension in KAGRA is approximately 13.5 m, the support structure is less than 1 m. Because a shorter support structure is more rigid, using this underground site is advantageous for constructing a tall mirror suspension.

### 2.4.2. Cryogenics

The thermal noise of mirrors and their suspensions is a fundamental noise source for laser interferometric GW detectors. A promising way to reduce thermal noise is to use cooling mirrors and their suspensions at cryogenic temperatures. Therefore, the mirrors and suspensions of the arm cavities in KAGRA were designed to be cooled to 20 K.

The KAGRA test-mass suspension, which is called a Type-A suspension, consists of nine stages: the upper five stages are at room temperature and the lower four stages are at cryogenic temperatures; they are called the Type-A tower and cryogenic payload, respectively. The cryogenic payload is suspended from the Type-A bottom filter, which is the bottom stage of the Type-A tower, and is stored in a cryostat. Figure 5 shows a schematic of the KAGRA cryogenic system. KAGRA's test masses are located at the bottom of a cryogenic payload in the bottom four stages of the main mirror suspension (the platform (PF), marionette (MN), intermediate mass (IM), and test mass (TM)). The MN, IM, and TM are surrounded by the corresponding recoil masses (marionette recoil mass, intermediate recoil mass, and recoil mass, respectively) to control the position and angle of the mirror. The mirror is made of monocrystalline sapphire, which has an extremely low mechanical loss of $10^{-8}$ at cryogenic temperatures [31], thereby reducing thermal noise. The sapphire mirror has a cylindrical shape with a diameter of 22 cm and a thickness of 15 cm. It is suspended using four sapphire fibers of 1.6 mm thickness and 350 mm length, which can extract heat from sapphire mirrors effectively owing to their high thermal conductivity at low temperatures [32]. In addition, sapphire mirrors have relatively low absorption at the laser wavelength (1064 nm) [33], resulting in less heat being generated during operation. The cryogenic payload is cooled through 6N (99.9999%) pure aluminum heat links [34], which yield high thermal conductivity while maintaining low stiffness, thus reducing the vibration transfer via heat links. In addition, they are connected to the MN stage to avoid direct coupling of the vibration via heat links to sapphire mirror motions.

The cryogenic payload is stored in two layers of radiation shields to avoid heating by thermal radiation. Sufficient cooling performance is obtained by cooling the inner and outer radiation shields to 8 K and 80 K, respectively. The payload, except for the sapphire parts and the inner side of the radiation shields, is coated with black plating called SOLBLACK and diamond-like carbon, thus effectively utilizing thermal radiation to cool the payload by obtaining a large emissivity. For continuous operation of the cooling system, KAGRA uses four one-watt cryocoolers to cool the cryogenic payload and radiation shields. Furthermore, two cryocoolers are used to cool the payload, and the others are used to cool the radiation shields. However, this generates large vibrations during operation and contaminates the sensitivity of the detector. Therefore, pulse-tube cryocoolers, which have very low vibrations [35], are used, and a heat-link vibration isolation system (HLVIS) is configured to further reduce the vibration transfer to the sapphire mirror via heat links.

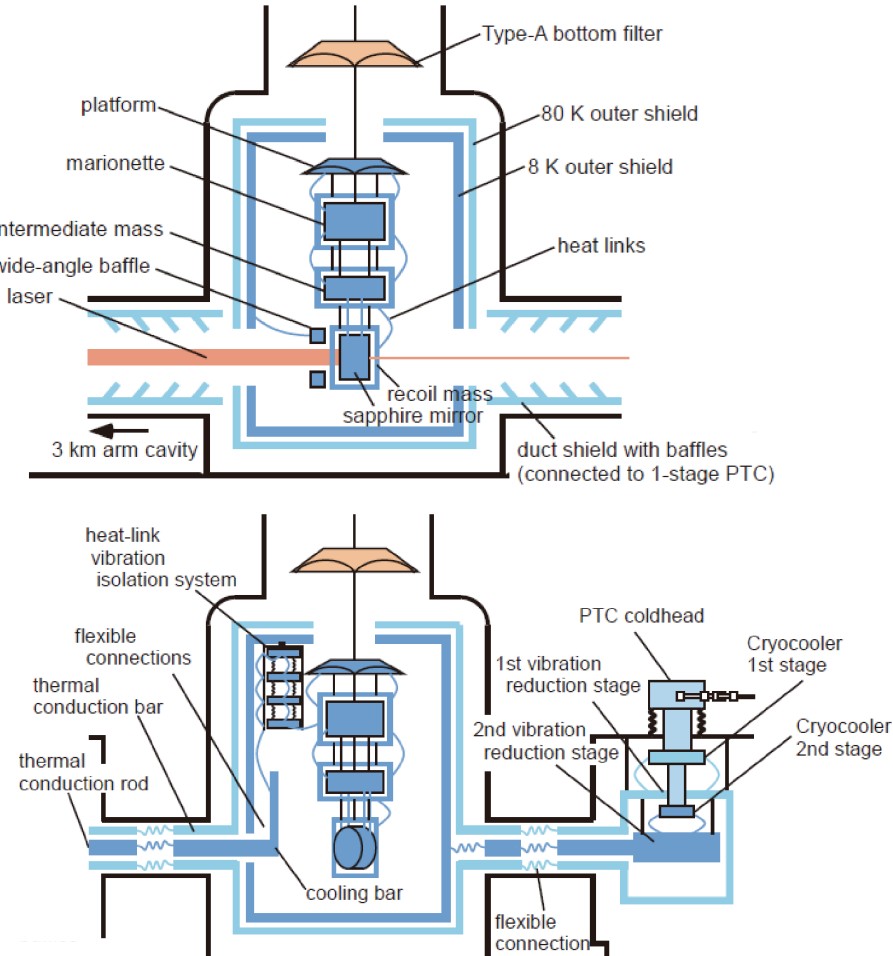

**Figure 5.** Schematic view of KAGRA's cryogenic system. A side view of the cryostat from the direction orthogonal to the arm (**top**) and another side view at an angle of 45° from the direction indicated in the left panel (**bottom**). Adapted with permission from Ref. [8], ©2021 Oxford University Press.

## 3. Recent Status

### 3.1. Detector Performance during the O3GK Observation Run

In April 2019, the Advanced LIGO detectors and the Virgo detector started their observation runs, which are called O3 [36]. In late March 2020, KAGRA was accepted to join O3 as a member of the scientific collaboration with LIGO and Virgo. However, LIGO and Virgo stopped their operations in March 2020 because of the COVID-19 pandemic. In this situation, GEO 600, which is the interferometric GW observatory in Germany with an arm length of 600 m [3], continued the observation, and GEO 600 and KAGRA started a joint observation run called O3GK. This was the first international observation run and a significant milestone for KAGRA.

One of the parameters describing the detector performance is the BNS inspiral range. This is the average distance at which the detector can detect typical binary neutron star mergers. The average BNS inspiral range of KAGRA in the O3GK was 660 kpc [37], although the best value was approximately 1 Mpc, which was recorded during the commissioning period. The interferometer configuration was PRFPMI, and the input power before PRM was 5 W, corresponding to an intracavity power of 50 kW.

The sensitivity with the dominant noise in the O3GK observation run is shown in Figure 6 [37]. As shown in the figure, most of the noise limiting the sensitivity was revealed. The sensitivity at frequencies lower than 100 Hz was limited by the suspension control noise. The details of the suspension control noise are described in Section 3.2.2.

From 100 to 400 Hz, acoustic noise was found to pollute the sensitivity. At the experimental site, many instruments generate sound, and the vibrations of the vacuum chamber caused by these sounds deteriorate the sensitivity. At the frequencies between 400 Hz and 2 kHz, the shot noise limits the sensitivity. Laser frequency noise is another noise that limits the sensitivity at high frequencies above 2 kHz. Although a frequency noise stabilization system was implemented, it was not optimized in the observation run. By optimizing the servo parameter, it will be reduced to lower than the shot noise.

Another important factor in gravitational wave detectors is the duty factor, which is the ratio of the time for which the interferometer is kept locked to the time of the observation period. Because the interferometer can lose its lock owing to external disturbances, such as earthquakes, a high duty factor is one of the major issues in constructing a GW detector. The duty factor of KAGRA during O3GK was 53%, whereas that of GEO 600 was 78% [37].

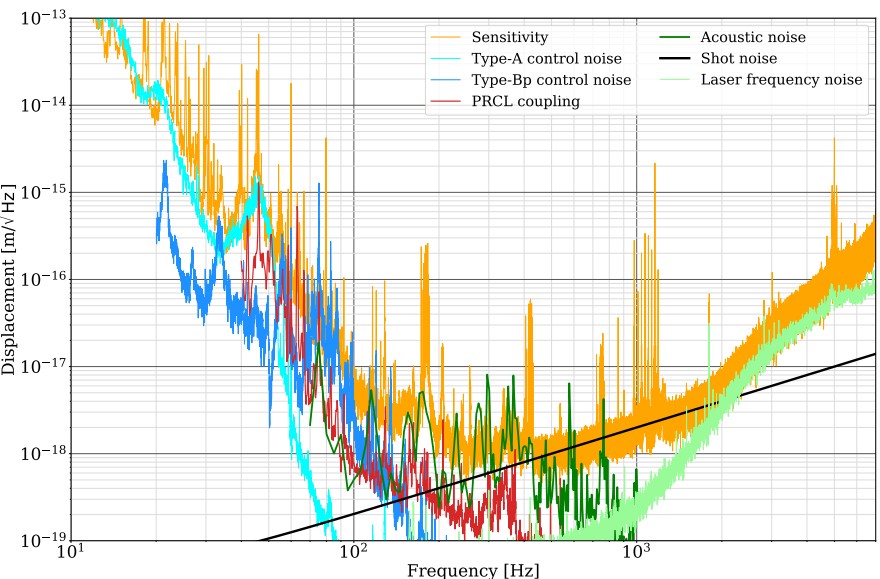

**Figure 6.** KAGRA's sensitivity with dominant noises. Suspension control noise (<100 Hz), acoustic noise (100 Hz~400 Hz), shot noise (400 Hz~2 kHz), and laser frequency noise (>2 kHz) limit the sensitivity. The average BNS inspiral range was 660 kpc for the O3GK period. Further details are shown in Ref. [37].

### 3.2. Toward the O4 Observation Run

The interferometer was sometimes difficult to lock during O3GK, especially on days with large microseismic motions (the details are explained in Section 4.1.1). KAGRA plans to upgrade the detector to improve the duty factor. In addition, because the sensitivity of KAGRA during O3GK was limited by two primary noise sources, quantum shot noise at high frequencies and suspension control noise at low frequencies, KAGRA also plans to improve the detector sensitivity for the international joint observation run starting from mid-December 2022 (O4) [38]. Furthermore, some technical difficulties inhibited the cooling of the sapphire mirrors during O3; therefore, several studies for achieving cryogenic operation have been performed for O4. KAGRA will start the O4 observation run with a sensitivity of over 1 Mpc and will work to improve the sensitivity toward the end of O4 by taking an observation break. In the following sections, the KAGRA upgrade plans and recent results on cryogenics are briefly reviewed.

### 3.2.1. Upgrade for Improving the Duty Factor

Seismic noise at low frequencies, especially microseismic noise, needs to be mitigated to improve the duty factor because it significantly affects the detector's stability. However, passive vibration isolation systems for such low frequencies are challenging; thus, active

vibration isolation using inertial sensors, such as a speed meter or accelerometer, is implemented to control the vibration isolation system for sapphire mirrors. Three accelerometers were installed on the main mirror suspensions for inertial control of the suspensions after O3GK. They can detect seismic motion at the level of $10^{-7}$ m/$\sqrt{\text{Hz}}$ at 0.2 Hz, which is approximately the same as the microseismic motion on a typical day and one order of magnitude smaller than that on a noisy day.

Another measure for improving detector stability is the installation of stronger actuators on the payload of the main mirror suspensions. Cryogenic payloads have coil–magnet actuators consisting of a coil and magnet, which move the suspension through electromagnetic force. However, sometimes the actuator was saturated during lock acquisition and observation, which triggered a lock loss in the interferometer. Therefore, a stronger actuator avoids saturation of the actuators and improves the duty factor. However, the electrical noise of analog circuits and DAC noise are coupled to sensitivity through actuators if the actuator efficiency is too large. Therefore, it is necessary to make the actuator efficiency as large as possible while maintaining a sufficiently low noise coupling in the observation band. Figure 7 shows the actuator efficiencies of the MN and IM stages. The new MN and IM actuators have efficiencies of 1.1 N/A and 55 mN/A, respectively, while the old MN and IM actuators had efficiencies of 0.47 N/A and 18 mN/A, respectively [39]. Based on the measured actuator efficiencies, the noise of analog electronics, and DAC noise, the noise caused by the MN and IM actuators can be estimated as $8.8 \times 10^{-20}$ m/$\sqrt{\text{Hz}}$ and $1.6 \times 10^{-19}$ m/$\sqrt{\text{Hz}}$ at 10 Hz, respectively, which are below the target sensitivity of KAGRA.

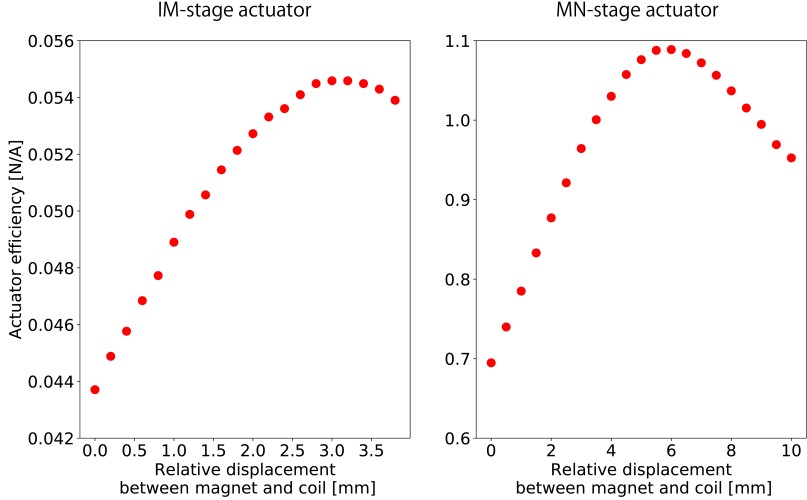

**Figure 7.** Actuator efficiencies of new coil–magnet actuators of the IM stage (**left**) and MN stage (**right**) as functions of the relative displacement between coils and magnets.

### 3.2.2. Upgrade for Improving Sensitivity

The sensitivity of KAGRA during O3GK was limited by the quantum shot noise in the high-frequency region. Thus, the replacement of the laser source with one of a higher power is planned for O4. Because quantum shot noise is proportional to $1/\sqrt{P}$, where $P$ is the intracavity power of an arm cavity, increasing the laser power can improve the sensitivity at high frequencies. A new laser source can output 60 W, while the maximum laser power of the current KAGRA laser source is 40 W. The performance of the new laser is currently under evaluation.

The sensitivity in the low-frequency region is limited by the suspension control noise. Especially below 50 Hz, it was contaminated by suspension control noise from the main mirror suspensions. Because the suspension system in a GW detector has a very complex structure, the suspensions have many resonant modes, which disturb the interferometer's operation. Therefore, local controls for damping these suspension res-

onances are necessary, but can contaminate the sensitivity. Reducing the sensor noise for damping controls is an effective way to mitigate control noise. The payload of a main mirror suspension has reflective photosensors for damping controls [39]; however, it has a large noise above 10 Hz, approximately $6 \times 10^{-9} \mathrm{m}/\sqrt{\mathrm{Hz}}$ for lateral motion and $4 \times 10^{-8} \mathrm{rad}/\sqrt{\mathrm{Hz}}$ for rotation. Therefore, optical levers [40] were installed at the MN and PF stages, which have approximately $2 \times 10^{-9} \mathrm{m}/\sqrt{\mathrm{Hz}}$ for lateral motion and $3 \times 10^{-10} \mathrm{rad}/\sqrt{\mathrm{Hz}}$ for rotation.

### 3.2.3. Recent Results on Cryogenics

Cooling mirrors for reducing thermal noise are a unique feature of KAGRA, adding certain difficulties related to cryogenics. One of them is molecular adsorption on the cryogenic mirror surface, which causes variations in the reflectivity of the mirrors and laser absorption in the molecular layers [41]. Because molecular layers of a few micrometers cause significant changes in the sensitivity of KAGRA, the mirrors need to be frequently warmed to desorb the molecules from the mirror surface. For this purpose, new heaters for the desorption of molecules were newly installed on the IM stage of the cryogenic payload to mitigate the downtime of observation. Owing to these new heaters, the downtime of the desorption process is expected to reduce from several weeks to a few days.

Four sapphire mirrors were cooled to cryogenic temperatures in 2019. Figure 8 shows an example of the cooling curve of the cryogenic system at the Y-end station from April to May 2019. Because thermal radiation is the dominant cooling path over 100 K, the inner radiation shield, mirror, marionette recoil mass, and HLVIS are cooled simultaneously. On the other hand, because conductive cooling is the dominant path of cooling below 100 K, cooling proceeds from the elements that are closer to the cryocoolers, in the order of HLVIS, marionette recoil mass, and mirror.

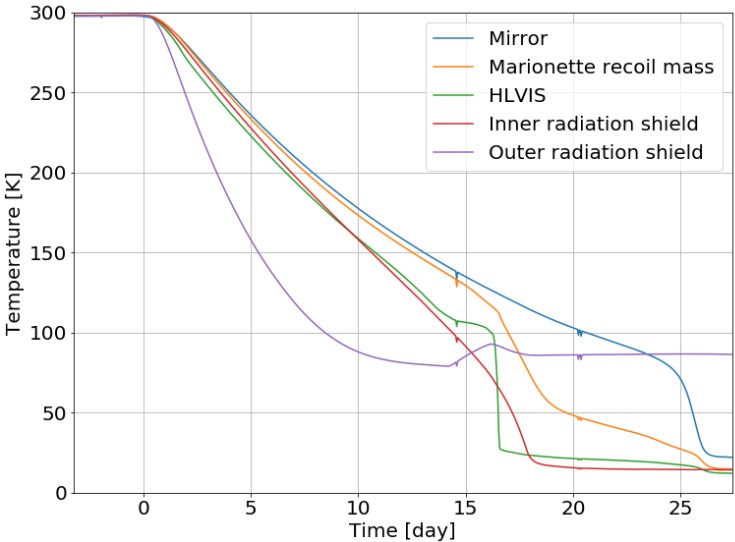

**Figure 8.** Example of the cooling curves of KAGRA's cryogenic system. The mirror reached 22 K at 27 days after the start of cooling. The cooling speed drastically changed below 100 K because of the increase in the thermal conductivity of the heat links and decrease in the specific heat. Each component is shown in Figure 5.

During cooling in 2019, KAGRA faced a more serious problem. The molecular adsorption during the initial cooling was much greater than expected, and visible frost was formed on the mirror surface. Figure 9 shows photographs of the mirror surface illuminated by a green laser, with and without frost on the mirror. As shown in Figure 9, a very thick frost was formed on the cryogenic mirror surface and caused significant scattering of the green beam. Once the thick frost was formed, the cavity finesse for the 1064 nm laser dropped to several hundred or less, while the finesse at room temperature was approximately 1500.

Therefore, a new cooling scheme to prevent frost formation was considered and tested from November 2020 to February 2021. The scheme involved cooling the mirrors step by step and trapping molecules not on the mirror surface but on the surface of the radiation shields. Owing to this new scheme, the KAGRA mirror was successfully cooled without any thick frost that could be visually inspected.

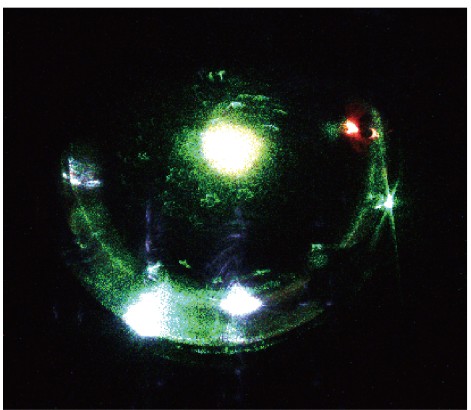 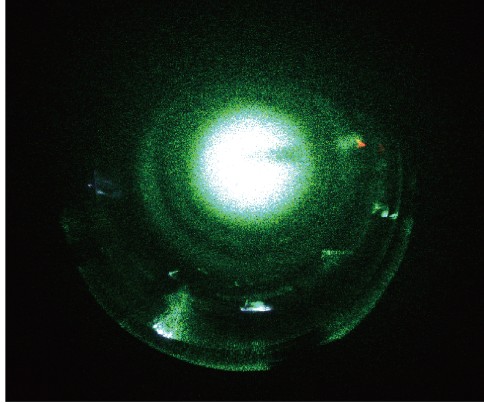

**Figure 9.** Images of the mirror illuminated with a green laser. (**Left**) Image at room temperature. The mirror surface is clean, and the green beam scattering is smaller than that at cryogenic temperature. (**Right**) Image at cryogenic temperature. The mirror surface is covered with thick frost, and the green beam is scattered on the surface.

## 4. Evaluations of the Underground Environment

A GW detector is a delicate system that is easily affected by environmental disturbances. The underground environment is expected to be quieter than that of the ground surface, and its actual evaluation is important not only for KAGRA, but also for further GW observatories, such as the ET. In this section, the current results of environmental studies in KAGRA are explained in terms of both their benefits and difficulties. An overview of the physical environmental monitoring system in KAGRA is described in Ref. [42].

### 4.1. Seismic Motion

4.1.1. Microseismic Motion

As discussed in Section 2.4.1, seismic motion at the frequency of the observational windows at the experimental site is significantly reduced compared with that on the ground surface. However, this reduction is inefficient at lower frequencies. Sometimes, microseismic motions caused by sea waves disrupt the control of suspensions and the interferometer. Japan is surrounded by two seas, the Pacific Ocean and the Sea of Japan, which exhibit different seasonal behaviors. Figure 10 (top) shows the wave height of the Pacific Ocean (Omaesaki in Shizuoka prefecture) and Sea of Japan (Wajima in Ishikawa prefecture) from July 2019 to June 2020 [43]. Figure 10 (bottom) shows the seismic spectrum at the KAGRA site on 12 October 2019 (green), December 2019 (cyan), and March 2020 (red). In the winter season, the waves of the Sea of Japan and the seismic motion at the KAGRA site became relatively larger. In summer and autumn, the level of the sea waves was usually low; however, sometimes it increased owing to a typhoon (for example, on 12 October 2019).

The relationship between the microseismic level and lock state of KAGRA's main interferometer during O3GK was studied [44]. Figure 11 shows the sea waves, seismic level at the KAGRA site, and lock state of the KAGRA interferometer with a focus on the O3GK term. The correlation between the sea waves and seismic level could be observed, and the interferometer could not be locked when the sea was rough (>2 m).

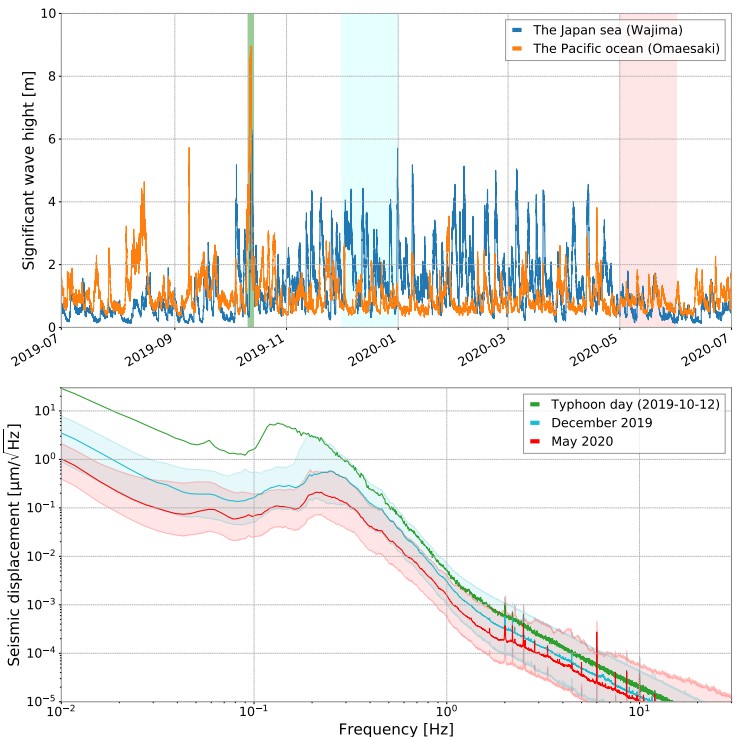

**Figure 10.** (**Top**): One-year data of the significant wave height in the Sea of Japan (Wajima) and Pacific Ocean (Omaesaki), opened in NOWPHAS [43]. (**Bottom**): Amplitude spectral density of the horizontal seismic displacement measured at the X-end of KAGRA. Each color corresponds to the period shown in the top graph. The solid lines represent the medians and the bands represent the 10th–90th percentiles during the one-month period.

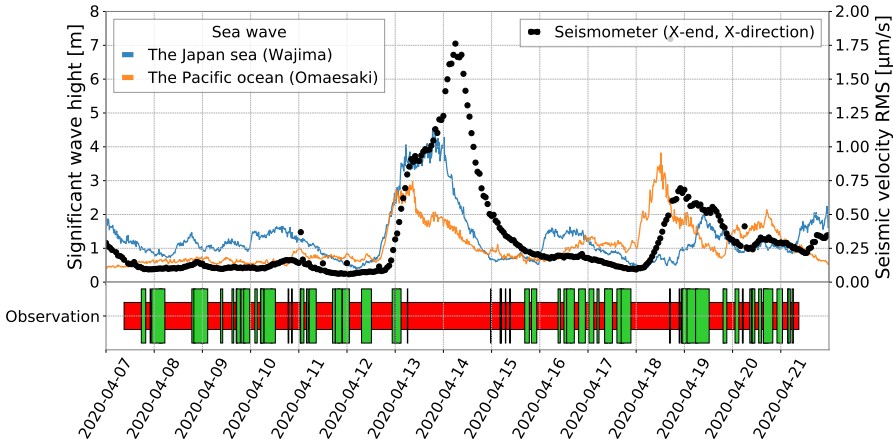

**Figure 11.** Comparison of the microseisms and the observation state of KAGRA during O3GK [44]. The blue and orange lines show the significant wave height in the Sea of Japan (Wajima) and Pacific Ocean (Omaesaki), respectively, opened in NOWPHAS [43]. The black markers are the RMS values of the seismic velocity in the 0.1–0.3 mHz band for every hour, measured at the X-end of KAGRA. The bottom bar graph shows the observation status of KAGRA during O3GK [37], with the science mode (green) and others (red, mainly the unlocked period).

### 4.1.2. Seismic Newtonian Noise

The motion of the mass around the experimental site induces a fluctuation in Newton's gravity and shakes the test-mass mirrors. This is called Newtonian noise (NN) [45]. It cannot be shielded against and is counted as fundamental noise.

Seismic motion is known to be the primary source of NN and has been intensively studied in GW detector research. For example, it is known that seismic Rayleigh waves propagating on a surface are reduced in underground facilities. Figure 12 shows the estimation of the NN caused by seismic body waves, seismic Rayleigh waves, and room acoustic waves for KAGRA [46]. All lines are significantly below the design sensitivity of KAGRA.

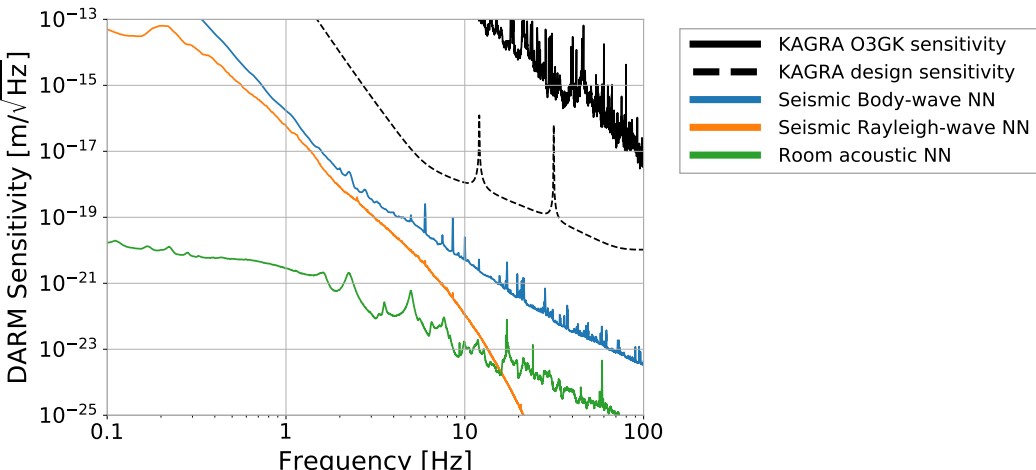

**Figure 12.** The estimated Newtonian noise from the seismic body waves (blue), seismic Rayleigh waves (orange), and acoustic fields in the experimental site (green) for KAGRA [46] compared with the O3GK sensitivity (black, solid) and design sensitivity (black, dashed) [37] of KAGRA.

*4.2. Acoustic Field*

The sound at the experimental site is considered environmental noise. Most of the main optics, such as test-mass mirrors or photodetectors for the GW detection port, are suspended in vacuum chambers and isolated from the acoustic field in the experimental room. However, the laser source and other auxiliary optics are not located in vacuum (this is not specified for the underground environment; however, the acoustic response of the KAGRA interferometer was carefully studied for the O3GK configuration [47]). Infrasound, which is a low-frequency sound that the human ear cannot detect, is also of interest because it causes the expansion and contraction of the arm tunnel [42]. The spectrum of the acoustic field in KAGRA's corner station (CS) and X-arm is compared with those of the Virgo central experimental building (CEB, a ground-surface facility) and Matra Gravitational and Geophysical Laboratory (MGGL, an underground facility) in Figure 13 under quiet conditions without human activity. The acoustic levels are similar for both datasets, and the difference between the underground and on-surface environments is not significant. Notably, however, the underground environment is quieter and more stable than the on-surface environment with respect to transient external acoustic disturbances, such as agricultural work or airplanes.

One unique aspect of the acoustic properties of KAGRA is that the reverberation time at the experimental site is much shorter than those of LIGO and Virgo; therefore, a transient sound decays quickly in KAGRA. This is because of the difference in the inner surfaces of the walls rather than the underground location. LIGO and Virgo have painted hard concrete walls, and they reflect the sound efficiently. On the other hand, the walls of KAGRA are coated with bubbling urethane and plastic paint, and they work as acoustic absorbers (Figure 14). A paper on the quantitative evaluation of this topic is in preparation.

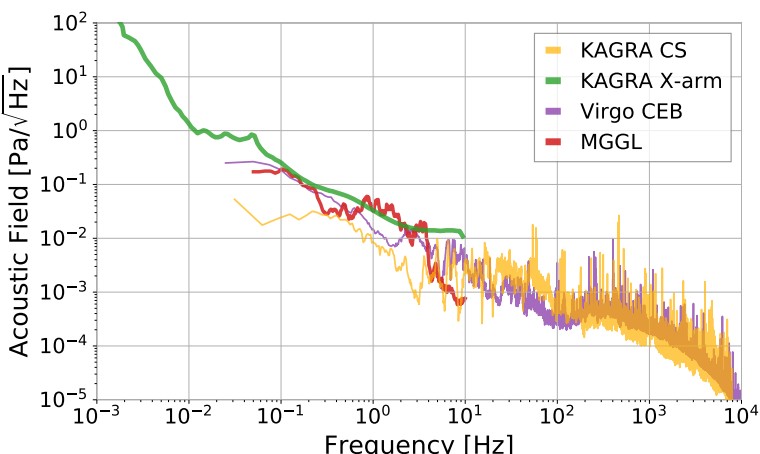

**Figure 13.** Comparison of the acoustic fields for the gravitational experimental sites: KAGRA CS [48], KAGRA X-arm [42], Virgo CEB [48], and MGGL [49].

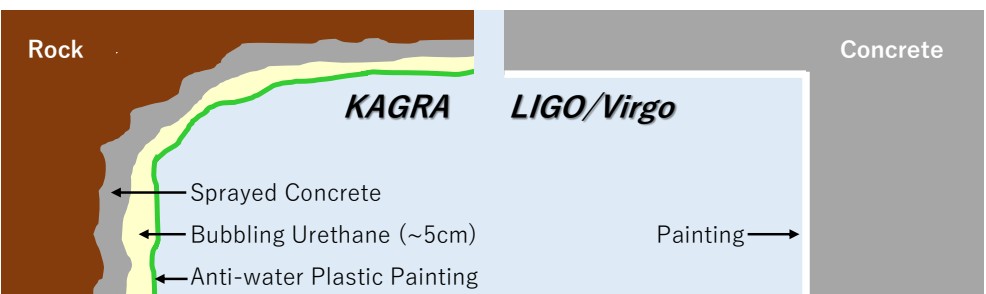

**Figure 14.** Schematic view of the walls in the experimental sites of KAGRA and LIGO/Virgo.

*4.3. Magnetic Field*

4.3.1. Magnetic Noise Estimation

A magnetic field can cause displacement noise through the force on the mirrors and/or sensing noise through the electronics.

The magnetic field in the cryostat was measured at the X-end. The results are presented in Figure 15 (top, red graph), and the peaks at 1 and 1.7 Hz correspond to the periods of the motors for the cryocoolers (1 and 0.6 s). The coupling functions in case 1 (measured for LIGO) and case 2 (measured for Virgo) were used to estimate the contribution of magnetic noise to the sensitivity of the DARM because it has not yet been measured for KAGRA. The approximated coupling function between the magnetic field and DARM displacement is written as

$$C(f) = \kappa \times \left(\frac{f}{f_0}\right)^{-\beta} \quad [\text{m/T}], \tag{1}$$

where $\kappa$, $\beta$, and $f_0$ ($\kappa = 8 \times 10^{-8}$ m/T, $\beta = 2.67$ in LIGO [50], $\kappa = 5.6 \times 10^{-8}$ m/T, $\beta = 3.3$ in Virgo [51], and $f_0 = 10$ Hz for both detectors) are the experimental parameters used to characterize the data. Using these coupling functions, the magnetic noise is projected onto the DARM sensitivity, as shown in Figure 15 (bottom). According to the estimation, magnetic noise will not contaminate the design sensitivity.

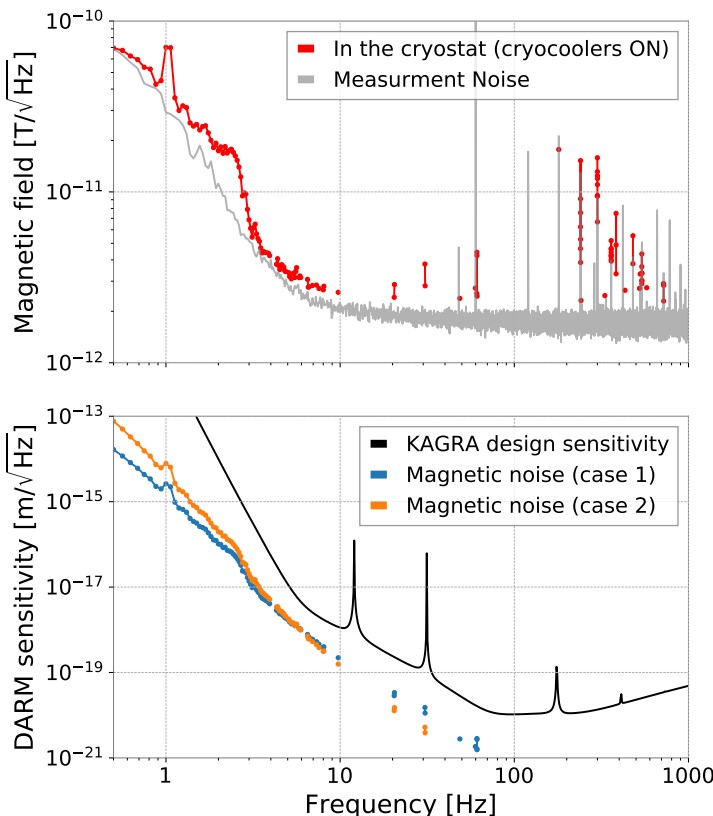

**Figure 15. (Top)** Magnetic field measured in a cryostat for the ETMX with cryocoolers. The gray line shows the measurement limit, including the sensor and ADC noises. **(Bottom)** Expected magnetic noise in the DARM sensitivity of KAGRA, calculated using the magnetic field in the top plot and the coupling functions evaluated for LIGO (blue) and Virgo (orange).

4.3.2. Schumann Resonance

The Schumann resonance is a global electromagnetic resonance with frequencies of 7.8, 14.1, 20.3 Hz, and so on, and an amplitude of approximately 1 pT/$\sqrt{\text{Hz}}$, which is generated and excited by lightning discharges in the cavity formed by the Earth's surface and ionosphere. Its contribution to a single GW detector as noise is expected to be smaller than that of the local magnetic field (for example, coming from power lines or electrical apparatuses); however, it has coherence between far-away points on Earth and is a common noise for the global GW observation network, especially when searching for stochastic background GWs [52,53].

Short-term Schumann resonance measurements were performed at the KAGRA experimental site during the construction phase [54,55]. Figure 16 shows the recent results of the measurements outside the tunnel and inside the KAGRA X-arm tunnel. The amplitude of the Schumann resonance (X-direction) is larger inside the tunnel than outside, a behavior that was also observed in the previous two measurements. More detailed studies, such as remeasurements and simulations, are ongoing to understand this behavior.

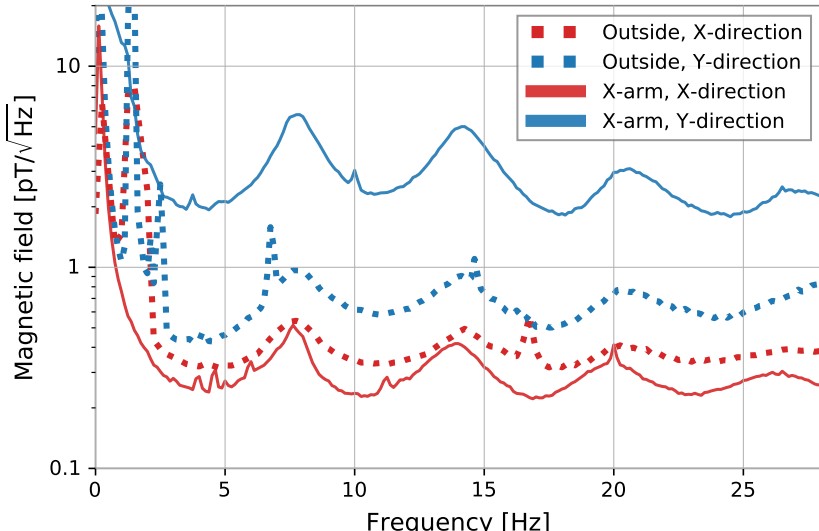

**Figure 16.** Schumann resonance of magnetic fields measured outside the tunnel (dashed) and inside the X-arm (solid). Red (blue) corresponds to the direction along the X(Y)-arm.

### 4.3.3. Transient Magnetic Noise from Lightning Strikes

Lightning strikes are well-known high-energy phenomena that emit transient magnetic noise to the atmosphere. When a lightning strike occurs close to KAGRA, a glitch event can be detected by both magnetometers and the GW channel of the main interferometer of KAGRA. Figure 17 shows an example of a lightning event [56]. This is the first evidence that a GW detector constructed in an underground facility is excited by lightning strikes in the atmosphere. This means that lightning is a background event of a burst-GW search, but it can be easily identified using the current system of environmental monitoring.

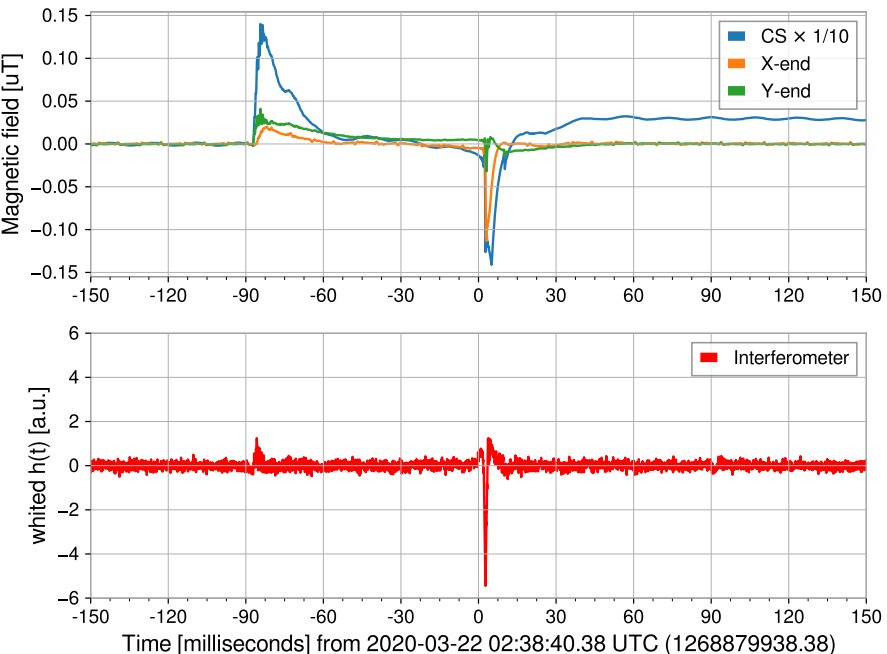

**Figure 17.** Time–series data of the magnetometers inside the KAGRA site (**top**) and KAGRA's main interferometer (whitened strain signal) (**bottom**) for nearby lightning. The origin of the horizontal axis (22 March 2022, 02:38:40.38) is the time of a lightning strike. For the magnetometers, the DC value of geomagnetism (∼50 µT) is subtracted. Reprinted with permission from Ref. [56], ©2021 IOP Publishing.

*4.4. Facility Issues*

Although underground facilities provide many benefits, they also present challenges, as explained in this paper. A limitation in space leads to poor extensibility, making it tough (and expensive) to build a filter cavity or long signal-recycling cavity in KAGRA.

In general, the temperature of an underground cave is stable throughout the seasons. However, at the KAGRA experimental site, the room temperature easily changes when the working status of the apparatus changes. The room-temperature change affects the control and alignment of the interferometer. For example, it causes a drift in the geometrical anti-spring filters for vibration isolation in the vertical direction. Typically, room temperature must be maintained within 0.2 °C.

Springwater is another practical issue for underground facilities. Because it can easily pass through the ground and reach the experimental area, it should be removed using pumps and waterways to avoid any accidents. These devices generate additional environmental noise for the GW detector, which must be mitigated. Newtonian noise coming from fluid in a drainage pipe is one possible noise, and its study in simulations is ongoing.

**5. Future Plan**

GW detection has opened up a new window in astronomy and astrophysics, with further developments expected in the future. Increasing the number of observable events by expanding the observation range and improving the estimation accuracy of source parameters with higher SNR signals are essential. A new detector in the same frequency band with a sensitivity one order of magnitude higher than that of the current GW detectors is under development. ET, which is proposed to be built in Europe, has an arm length of 10 km, mirrors cooled to cryogenic temperatures [9], and underground construction. In addition, the LIGO project proposes the construction of a detector named Cosmic Explorer with an arm length of 40 km, whose sensitivity will be an order of magnitude higher than that of the Advanced LIGO detectors [10]. Another approach to advancing GW astronomy is launching GW observatories into space to observe GWs in lower frequency bands, such as with LISA [57], DECIGO [58], and TianQuin [59].

Although these attempts are being vigorously pursued, construction may take more than a decade. In the meantime, further improvement in the sensitivity of existing detectors is essential for the continuous development of GW astronomy. Advanced LIGO plus (A+) and Advanced Virgo Plus (AdV+) [60], which are upgraded detectors of Advanced LIGO and Advanced Virgo, are planned, and each is expected to improve the sensitivity by a factor of two compared to the current design sensitivity. In addition, in KAGRA, various proposals have been discussed [61]. In this section, some of the proposed plans are introduced.

One of the proposals involves improvement of the sapphire mirror. By making the mirror larger and heavier, domination of the thermal noise of the suspension and the quantum radiation pressure noise over the sensitivity in the low-frequency region will be reduced. Moreover, by increasing the beam size, the coating's thermal noise is expected to be reduced. The currently used sapphire mirror is 22 cm in diameter, 15 cm thick, and weighs 23 kg, which was the largest size that could be made when it was constructed. On the other hand, as a result of research and development, it is expected that a 100 kg sapphire crystal with a diameter of 36 cm and thickness of 25 cm will be created in a few years.

According to the quantum uncertainty principle, the product of the amplitude and phase fluctuations of light has a finite magnitude, and the two fluctuations cannot be simultaneously reduced to zero. A quantum squeezing technique is used to reduce only one of these fluctuations without violating the quantum uncertainty principle by sacrificing the other fluctuation. In Advanced LIGO and Advanced Virgo, reduction of the quantum shot noise, which is caused by quantum phase fluctuations, has been successful [62,63]. However, the intensity fluctuation becomes larger under the squeezed condition in the phase fluctuation, and the radiation pressure noise, which is caused by the quantum

amplitude fluctuation, becomes larger. Therefore, a new technique, called frequency-dependent squeezing, is currently under development. This technique uses an optical resonator called a filter cavity to add frequency dependence to the squeezing process. While the quantum phase fluctuation is squeezed in the high-frequency region, where quantum shot noise is dominant, the quantum amplitude fluctuation is squeezed in the low-frequency region, where quantum radiation pressure noise is dominant. Thus, effective quantum noise reduction across the entire bandwidth is achieved. The frequency-dependent squeezing technique was demonstrated by the MIT group in the U.S. [64] and the NAOJ group in Japan [65–67] and will be installed in Advanced LIGO and Advanced Virgo before O4. It is anticipated that this technique will be adopted by KAGRA.

The improved sensitivity obtained by combining these techniques is shown in Figure 18. KAGRA's sensitivity can be improved to the same level as that of A+ and AdV+. Because the mirrors of KAGRA are cooled to cryogenic temperatures, it is difficult to reduce the noise by increasing the laser power, as is planned for A+ and AdV+. Increasing the laser power makes it difficult to cool the mirror to a cryogenic temperature owing to the heat caused by the absorption of the mirror. Because the quantum and thermal noise can be reduced without increasing the laser power, the frequency-dependent squeezing technique and the use of a larger sapphire mirror are the best strategies for improving the sensitivity of KAGRA. This is also the case for the next generation of GW detectors, for some of which cryogenic mirrors are planned, and the improved sensitivity of KAGRA can serve as a case study for them.

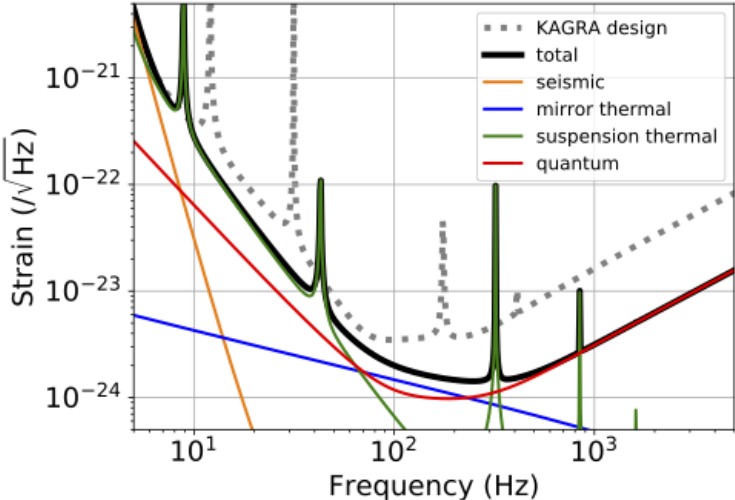

**Figure 18.** Estimated sensitivity curve after the modifications described in this section. Adapted with permission from Ref. [61], ©2020 American Physical Society.

## 6. Conclusions

In this article, the current status and future upgrades of KAGRA were reviewed. The first joint observation run in 2020 was a major milestone for KAGRA; however, further upgrades are necessary in order to contribute to gravitational wave astrophysics. KAGRA's key features, the suspensions and cryogenic system, are being upgraded, which is significantly important for the next observation run. Evaluations of the unique underground environment of KAGRA have also progressed, and some topics are progressing ahead of LIGO and Virgo. GW detection with KAGRA is vital for promoting GW astronomy and will also serve as the basis for introducing new technologies in future GW observatories.

**Author Contributions:** Corresponding authors wrote all the manuscript. M.N. summarized IFO configuration, design sensitivity, and future plans. T.U. described on suspensions, cryogenic systems, and upgrade plans for O4. T.W. wrote on the underground environments. All authors were contributing to KAGRA project in 2020 and agreed to publish the manuscript. All authors have read and agreed to the published version of the manuscript.

**Funding:** This work was funded by MEXT, JSPS Leading-edge Research Infrastructure Program, JSPS Grant-in-Aid for Specially Promoted Research 26000005, JSPS Grant-in-Aid for Scientific Research on Innovative Areas 2905: JP17H06358, JP17H06361, and JP17H06364, JSPS Core-to-Core Program A. Advanced Research Networks, JSPS Grant-in-Aid for Scientific Research (S) 17H06133 and 20H05639, JSPS Grant-in-Aid for Transformative Research Areas (A) 20A203: JP20H05854, the joint research program of the Institute for Cosmic Ray Research, University of Tokyo, National Research Foundation (NRF), and Computing Infrastructure Project of KISTI-GSDC in Korea, Academia Sinica (AS), AS Grid Center (ASGC), and the Ministry of Science and Technology (MoST) in Taiwan under grants including AS-CDA-105-M06.

**Institutional Review Board Statement:** Not applicable.

**Informed Consent Statement:** Not applicable.

**Data Availability Statement:** Not applicable.

**Acknowledgments:** M. Nakano, T. Ushiba, and T. Washimi are grateful to Gabriele Vajente for inviting them to write the present manuscript for publication in Galaxies. This work was supported by Advanced Technology Center (ATC) of NAOJ, Mechanical Engineering Center of KEK, the LIGO project, and the Virgo project. Help in the study of the underground environment was provided by the Virgo/ET members, especially F. Paoletti, I. Fiori, J. Harms, and F. Badaracco.

**Conflicts of Interest:** The authors declare no conflict of interest.

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
