# Peer review of "The Current Status and Future Prospects of KAGRA, the Large-Scale Cryogenic Gravitational Wave Telescope Built in the Kamioka Underground"

_galaxies, doi:10.3390/galaxies10030063_

Round 1

Reviewer 1 Report

The paper presents the status of the KAGRA interferometer. As underlined in the text, these studies are of great interest for the scientific community of GW, mostly for the 3rd generation detectors. I whish some arguments were better clarified and exposed, as suggested in the attached report.

Author Response

Thank you so much for all of your kind comments. This is the summary of our answers to your comments. Responses to comments regarding simple English mistakes, paraphrases, etc. are not listed here, but unless otherwise noted, we have fixed them as commented.

-----------------------------------------------

Line 31: is only the observatory achieved the direct GW detection → try to rephrase this sentence, it’s not clear what you mean  

We rephrase it.  

Line 125: GENERAL COMMENT: It would be worth to add some formulas to better explain the noise sources, especially when talking about quantum shot noise. Moreover, I think the whole paragraph should be re-written better taking into account the different noise sources (see comments below).

We discussed and think that the exact formula is too complicated and out of the scope of this manuscript. We added several sentences describing the general characteristics of the noise, and we want readers to refer to the cited papers for further details.
And, we agree this paragraph is not kind enough for the reader in the sense of the difference between fundamental noise and technical noise, so we modified several sentences and tried to clarify this point.

Line 127: swinging only about 3 × 10^(−18) m in the case of KAGRA → does the amplitude o the gravitational wave depend on the detector? I would remove “in the case of KAGRA”. Also, why exactly 3× 10^(−18) m and not only the order of magnitude?

You are right. We removed the sentence ‘in the case of KAGRA’ and changed the number to “the order of 10^(−18) m”

Line 132: This type of noise is represented by seismic noise and thermal noises. Why control noise is not mentioned here? Moreover, at low frequency, radiation pressure noise, which is related to the quantum shot noise, produce an effective shake of the mirrors. Please try to clarify better these points.

About the radiation pressure noise, we rephrased the sentence about it.
About control noise, we did not list it because it is one of the technical noises.

Line 139: As in the previous comment, the radiation pressure noise seems to be dominant also at low frequency.

Added the radiation pressure noise in the list.

Line 141: is 1μm the RMS value all over the band below 5Hz? Otherwise, could you better specify the frequency band?

Yes, the RMS value is below 5Hz.  

Line 150: It would be nice to have a formula here The description on the frequency dependency of mirror thermal noise and suspension thermal noise.

Because the detail calculation of KAGRA thermal noise have already been well summarized in the ref. [27] and too complicated to write down here, we would like to leave it to the reference. 

Line 154-166: could you better specify the contribution of mirror thermal noise and suspension thermal noise, and in which bands they are dominant?

The description of mirror thermal noise and suspension thermal noise are added.

Line 162: The other approach→which other? 

The sentences are rephrased to clarify the corresponding contrasts.

Line 168: and it is another noise caused by quantum fluctuations of light. → Please, try to better clarify the role of quantum noise all over the paragraph, as radiation pressure at low frequency and shot noise at high frequency.

We modified the whole paragraphs about quantum noise, and try to clarify more about the quantum noise.

Line 175-184: maybe a difference should be highlighted between technical noises (such as laser frequency or amplitude noise), fundamental noises (like the ones mentioned above) and environmental noises, which can be spotted and mitigated through noise hunting. Please clarify better the difference.

We tried to highlight by adding the sentence from “The magnitude of fundamental noise….”

Line 189-194: two main reasons→these two main reasons can be classified as sensitivity degradation and duty cycle. Mirrors angular motion can induce both: it can induce lock loss if its amplitude is too wide and can degrade the sensitivity in case of angular-to-longitudinal coupling.

A description of angular-to-longitudinal coupling is added.

Line 216: which is the frequency band where you define the seismic motion as microseism? 

The frequency band is added. 

Line 220-227: please, clarify better which is the effect seen in Virgo which is not seen in KAGRA thanks to the different configuration.

Whole paragraph was reconsidered and rephrased. We don’t have concrete result on the suspension performance difference due to the structural differences. In addition, the main topic of this subsection is not a comparison with the other GW detectors but the general advantages of using underground site. Therefore, we don’t state the performance comparison here, and show a superattenuator in Virgo for just an example of a successful tall suspension constructed on the surface of the earth. 

Line 292: could you replace (also in the rest of the test) duty factor with duty cycle? 

Thanks for your comment, but we would like to use duty factor, since previous discussions between LIGO and KAGRA concluded that duty factor is a more appropriate word than duty cycle.   Figure 11 (caption): missing reference in the last line Added

Line 413: being underground reduces or not the effect of NN on KAGRA? It’s not clear from this paragraph.

The NN for KAGRA if it is located on the surface was not evaluated and is impossible to mention here.

Line 470: why is the inside noise level higher than the outside one? Is it expected? And why is it different along X and Y direction? Could you add these comments to the text?   

It is not understood yet. So we added the sentence "More detailed studies such as re-measurements or simulations to understand this behavior are ongoing." 

Best,
Takafumi, Tatsuki, and Masayuki

Reviewer 2 Report

This paper is a good overview of the status of KAGRA and the technological challenges connected with cryogenic development and underground locations.

Here are my thoughts and comments.

Page 3, Line 37- 38:Thus, the current GW detectors have a large km-scale arm; nevertheless, the typical GWs from the target
source cause only an arm length fluctuation of 10−21 m. 
 Is this number in a unit of strain?

page 9, Line 221- 227:
 "An additional advantage of underground sites is the ability to construct a huge suspension without tall support structures. For instance, Virgo
 constructed a tall suspension called a superattenuator [29] for efficient
 seismic isolation at low frequencies; however, they require a huge support structure that has low-frequency resonance and introduces large motion into the suspensions. On the other hand, KAGRA has a second floor in the mine and a test mass suspension that is supported from the second floor.
Therefore, even though the total height of the suspension is approximately 13.5 m,
 the suspension support is very rigid and reduces the undesirable resonance
at low frequencies"

what do you mean with the low-frequency resonance of the structure introducing large motion into the suspension? Where did you find this information?
The structure has inner frequencies around a few Hz where the passive attenuation of the Superattenuator is effective and this should filter any external disturbance. Anyway at  the moment there is no evidence that the structure introduces motion into the suspension.

Page 17, line 414-419 I understand what do you mean, but it is not clear where you define the NN. You should rephrase it.

Figure 15: The expected magnetic noise in DARM sensitivity, the projection shown in case 1 and case 2
are calculated Whit LIGO TF and Virgo TF, respectively. Is my understanding correct?

Author Response

Thank you so much for all of your kind comments. This is the summary of our answers to your comments. 

-----------------------------------------------

Page 3, Line 37- 38:Thus, the current GW detectors have a large km-scale arm; nevertheless, the typical GWs from the target
source cause only an arm length fluctuation of 10−21 m. 
 Is this number in a unit of strain?

We are sorry, the number was strain although the unit shows in length. We fixed it.

page 9, Line 221- 227:
 "An additional advantage of underground sites is the ability to construct a huge suspension without tall support structures. For instance, Virgo
 constructed a tall suspension called a superattenuator [29] for efficient
 seismic isolation at low frequencies; however, they require a huge support structure that has low-frequency resonance and introduces large motion into the suspensions. On the other hand, KAGRA has a second floor in the mine and a test mass suspension that is supported from the second floor.
Therefore, even though the total height of the suspension is approximately 13.5 m,  the suspension support is very rigid and reduces the undesirable resonance
at low frequencies" what do you mean with the low-frequency resonance of the structure introducing large motion into the suspension? Where did you find this information?
The structure has inner frequencies around a few Hz where the passive attenuation of the Superattenuator is effective and this should filter any external disturbance. Anyway at  the moment there is no evidence that the structure introduces motion into the suspension.

The whole paragraph is reconsidered and rephrased. The main topic of this subsection is not a comparison with the other GW detectors but the general advantages of using underground site. So, we show a superattenuator in Virgo here for just an example of a successful tall suspension constructed on the surface of the earth.

Page 17, line 414-419 I understand what do you mean, but it is not clear where you define the NN. You should rephrase it.

We modified the sentences to be clear and added the reference.   

Figure 15: The expected magnetic noise in DARM sensitivity, the projection shown in case 1 and case 2 are calculated Whit LIGO TF and Virgo TF, respectively. Is my understanding correct?

Yes, that's right. we added the information that each color are corresponding to the LIGO TF and Virgo TF in the caption 

Best regards,
Takafumi, Tatsuki, and Masayuki

Reviewer 3 Report

The paper describes the current status of the KAGRA detector, its main noises and problems as well as future plans for improving its sensitivity. It is an interesting paper for the gravitational wave community in light of the next observing run foreseen in 2022. It is also interesting from the point of view of the construction of ET, since this paper highlights a few important problems faced by KAGRA that will need to be addressed for ET too. I recommend the publication in the Galaxies journal.

However, before publication I recommend some improvements:

  1. I recommend a deep revision of the grammar and the language. As it is now, it is very hard to follow what there is written.

  2. In the abstract it is mentioned GEO600, but I didn’t see it mentioned in the introduction along with the others detectors. I suggest to add a few lines.

  3. Being a paper focused on the detector instrument I suggest to focus the introduction more on the instrument rather than what a gravitational wave is. I suggest then to mention the various noises affecting the detectors and putting references about them.

  4. Lines 24-25, yes, but from that sentence it could seem that current GW interferometers are capable of detecting this primordial signal. Which is not. I suggest to specify better.

  5. Lines 26-28: “Because the interaction between GWs and mass…” which mass? The source, the test mass…? To someone outside the field this will not be understandable.

  6. Line 29: 10^-18-m --> 10^-18 m

  7. Line 30-31: I don’t understand what there is written here.

  8. Line 37. I think that the sentence “longer arm interferometer is more sensitive to the GW interaction. Thus, the current GW detectors have a large km-scale arm” is inaccurate. GW detectors are effective not only because they have km long arms, but in particular because there are Fabry Perot cavities. The km length alone would not be effective for GW detection. I suggest to improve this line.

  9. Line 39 “arm length fluctuation of 10^-21 m”. It is not consistent with line 29. If you mean that the strain is 10^-21 (it’s dimentionless), then yes, the fluctuation in arm lengths of the order of the km is 10^-18 m.

  10. Lines 52-53: cryogenic mirrors are not used to construct the interferometer. This sentence does not make sense. I suggest something like: “… and makes use of cryogenic mirrors to improve the sensitivity of the detector lowering its thermal noise”

  11. Line 61 the Authors mention the mountain Ikenoyama, then they talk about Tateyama in Line 70. It is a bit confusing for who does not know the local geography. I suggest to dd a few lines explaining the what is what.

  12. Line 84. Why it is difficult? I would replace the sentence saying something like: “3 km arms do not suffice to reach the required sensitivity for detecting a GW, therefore Fabry Perot cavities are introduced.”

  13. Line 91: ETM acronym is not specified

  14. Line 97: again à back

  15. Lines 98-100. Which output signal? The GW signal is not proportional to the laser power. I would instead write that the shot noise, limiting the detector at higher frequencies, is proportional to the inverse of the square root of the power, so higher power means less noise, then higher signal to noise ratio at higher frequencies.

  16. Line 103: In reference [22] cited here, there is written: “By using the principles of coupled cavities, it is possible to increase the finesse of the arm cavities beyond that given by the storage-time limit, while keeping the storage-time for the signal sidebands consistent with the desired detection bandwidth. This is accomplished by adding a signal extraction mirror (M3 in fig. 1) to the standard configuration, the purpose of which is to decrease the storage-time for the signal sidebands. This is not to be confused with a “signal recycling mirror” [4] which was proposed as a means of increasing the signal sideband storage-time” In the Author’s text instead, the SRM and the RSE seems to be the same thing. I suggest to improve this point. I also found another reference about this topic: doi.org/10.1364/AO.42.001283 where it is explained the difference between the two things.

  17. Line 127: What do the Authors mean with swinging only about 3 10^-18 m? This point is obscure.

  18. Line 147: The thermal noise depicted in Fig 3 seems to affect the sensitivity beyond 80 Hz (up to 140 Hz). Moreover, in Fig 3 it is not mentioned the radiation pressure since it is embedded in the quantum noise curve. I suggest to write somewhere that they are the same thing.

  19. Line 174: here reference [24] is used referred to the design sensitivity curve, while in the caption of Fig3 is used ref [25]. It’s not really consistent. I suggest to insert one or the other.

  20. Section 2.4 is made by only one sentence. I suggest to incorporate subsections 2.4.1 and 2.4.2 in one whole section.

  21. Line 209: TAMA is never mentioned before. References are missing.

  22. Line 264: pules --> pulse

  23. Sec 3: in figure 6 is shown that the laser frequency noise is limiting at high frequencies. However, I couldn’t find any mention of it in the text. I suggest adding a few sentences about this too.

  24. Subsect 3.2.3: Resent --> Recent

  25. Line 367: ETMY only defined in caption of fig 2. Not clear

  26. Line 414: citation needed when the Newtonian noise is mentioned. I suggest https://link.springer.com/article/10.1007%2Fs41114-019-0022-2#citeas

  27. Lines 423-425: “Because the target frequency (10 Hz and above) for the GW observation of KAGRA overlaps with the sensitivity of the human ear (20 Hz–20 kHz), the audible sound in

    the experimental site is one of the environmental noises” The acoustic noise is due to the pressure generated by the sound, which can be at frequencies even below the human capability of hearing that. The two things (human ear and GW sensitivity to acoustic noise) are not related. I suggest to remove this sentence.

  28. Fig 12 and 16: the legend is not clear: dashed and continuous lines get confused.

  29. Lines 531-32 Also Virgo succeeded in the squeezing. Please add this reference too: https://journals.aps.org/prl/abstract/10.1103/PhysRevLett.123.231108

Author Response

Thank you for all of your kind comments. This is the summary of our answers to your comments. Responses to comments regarding simple English mistakes, paraphrases, etc. are not listed here, but unless otherwise noted, we have fixed them as commented.

-----------------------------------------------

I recommend a deep revision of the grammar and the language. As it is now, it is very hard to follow what there is written.

We are going to send it through the English proofreading process again as soon as the review has been completed. 

In the abstract it is mentioned GEO600, but I didn’t see it mentioned in the introduction along with the others detectors. I suggest to add a few lines.

We added the sentence about the GEO600 in the introduction section  

Being a paper focused on the detector instrument I suggest to focus the introduction more on the instrument rather than what a gravitational wave is. I suggest then to mention the various noises affecting the detectors and putting references about them.

Thanks for your comment, but we feel that such details of noise are too technical for the concept of this special issue. So, we concluded that it is better not to mention it.

Lines 24-25, yes, but from that sentence it could seem that current GW interferometers are capable of detecting this primordial signal. Which is not. I suggest to specify better.

We modified it. 

Lines 26-28: “Because the interaction between GWs and mass…” which mass? The source, the test mass…? To someone outside the field this will not be understandable.

 We rephrased it.

Line 30-31: I don’t understand what there is written here.

We rephrase it. 

Line 37. I think that the sentence “longer arm interferometer is more sensitive to the GW interaction. Thus, the current GW detectors have a large km-scale arm” is inaccurate. GW detectors are effective not only because they have km long arms, but in particular because there are Fabry Perot cavities. The km length alone would not be effective for GW detection. I suggest to improve this line.

We rephrase it 

Line 39 “arm length fluctuation of 10^-21 m”. It is not consistent with line 29. If you mean that the strain is 10^-21 (it’s dimentionless), then yes, the fluctuation in arm lengths of the order of the km is 10^-18 m.

We are sorry, you are right. We fixed it

Lines 52-53: cryogenic mirrors are not used to construct the interferometer. This sentence does not make sense. I suggest something like: “… and makes use of cryogenic mirrors to improve the sensitivity of the detector lowering its thermal noise” 

We modified the sentence as you suggested. 

Line 61 the Authors mention the mountain Ikenoyama, then they talk about Tateyama in Line 70. It is a bit confusing for who does not know the local geography. I suggest to dd a few lines explaining the what is what.

We added the map in the Fig.1. 

Line 84. Why it is difficult? I would replace the sentence saying something like: “3 km arms do not suffice to reach the required sensitivity for detecting a GW, therefore Fabry Perot cavities are introduced.”

We rephrased it. 

Line 91: ETM acronym is not specified

It is specified in line 83 in the new manuscript.

Line 97: again à back

We are sorry but we could not get your comment. You are suggesting rephrasing it?

Lines 98-100. Which output signal? The GW signal is not proportional to the laser power. I would instead write that the shot noise, limiting the detector at higher frequencies, is proportional to the inverse of the square root of the power, so higher power means less noise, then higher signal to noise ratio at higher frequencies.

We agree with you that this sentence is misleading, so we modified it. (However, we think the GW signal is proportional to the laser power, while the shot noise is proportional to the square root of the laser power. And as the result, the SNR to the shot noise is proportional to the inverse of the square root of the power. Is this disagree with your comment?)

Line 103: In reference [22] cited here, there is written: “By using the principles of coupled cavities, it is possible to increase the finesse of the arm cavities beyond that given by the storage-time limit, while keeping the storage-time for the signal sidebands consistent with the desired detection bandwidth. This is accomplished by adding a signal extraction mirror (M3 in fig. 1) to the standard configuration, the purpose of which is to decrease the storage-time for the signal sidebands. This is not to be confused with a “signal recycling mirror” [4] which was proposed as a means of increasing the signal sideband storage-time” In the Author’s text instead, the SRM and the RSE seems to be the same thing. I suggest to improve this point. I also found another reference about this topic: doi.org/10.1364/AO.42.001283 where it is explained the difference between the two things.

We think the concept of RSE is well described in this paragraph, but the name of the signal recycling mirror/cavity is misleading. It should be called signal extraction mirror/cavity in RSE interferometer, but due to the historical background, we mostly call that mirror as the signal recycling mirror/cavity which is so confusing. We added this background in the footnote. 

Line 127: What do the Authors mean with swinging only about 3 10^-18 m? This point is obscure.

We rephrased it. 

Line 147: The thermal noise depicted in Fig 3 seems to affect the sensitivity beyond 80 Hz (up to 140 Hz). Moreover, in Fig 3 it is not mentioned the radiation pressure since it is embedded in the quantum noise curve. I suggest to write somewhere that they are the same thing.

That's right. We modified whole paragraph, and we think it fixed this point.

Line 174: here reference [24] is used referred to the design sensitivity curve, while in the caption of Fig3 is used ref [25]. It’s not really consistent. I suggest to insert one or the other.

We are sorry, that's right. We fixed it. 

Section 2.4 is made by only one sentence. I suggest to incorporate subsections 2.4.1 and 2.4.2 in one whole section.

Some description on the following subsections are added instead of incorporating subsections into one section because each subsection is not so short and we felt some difficulties to merge them into one section.

Line 209: TAMA is never mentioned before. References are missing.

We added the reference.

Sec 3: in figure 6 is shown that the laser frequency noise is limiting at high frequencies. However, I couldn’t find any mention of it in the text. I suggest adding a few sentences about this too.

It was mentioned, but we guess the order is not good. We changed the order to mention each noise by the frequency region.

Line 367: ETMY only defined in caption of fig 2. Not clear

Because cooling curves include not only a test mass at Y-end station but also the surrounding system like radiation shields, “ETMY” was changed to “cryogenic system at Y-end station”. Also, to clarify each component written in the figure, fig.5 was referred in the caption of the figure. 

Line 414: citation needed when the Newtonian noise is mentioned. I suggest https://link.springer.com/article/10.1007%2Fs41114-019-0022-2#citeas

We added it into the reference.

Lines 423-425: “Because the target frequency (10 Hz and above) for the GW observation of KAGRA overlaps with the sensitivity of the human ear (20 Hz–20 kHz), the audible sound inthe experimental site is one of the environmental noises” The acoustic noise is due to the pressure generated by the sound, which can be at frequencies even below the human capability of hearing that. The two things (human ear and GW sensitivity to acoustic noise) are not related. I suggest to remove this sentence.

We removed the sentence. 

Fig 12 and 16: the legend is not clear: dashed and continuous lines get confused.

They are fixed.

Lines 531-32 Also Virgo succeeded in the squeezing. Please add this reference too: https://journals.aps.org/prl/abstract/10.1103/PhysRevLett.123.231108 

We added the reference.

Best regards,
Takafumi, Tatsuki, and Masayuki

Round 2

Reviewer 1 Report

I would recommend some minor revision that I suggested in the attached .pdf file. 

Author Response

Thank you for reviewing our manuscript again. The followings are the replies to your comments. For the other comments not mentioned here, we have modified them as your suggestion.

l280 - l282: "not very clear sentence. Please rephrase."
l283 - l285: "the word "suspension" is over-used. Please consider rephrasing also this sentence, which is not very clear either."

We rephrased these sentences as follows:
"For achieving a long suspension on the ground, it is necessary to construct a tall support structure and hang the mirror from its top like superattenuator in Virgo [31]. In contrast, it is unnecessary to build a tall support in KAGRA because KAGRA excavated a two-story tunnel and the mirrors can be hanged from the second floor.".

Best,
Takafumi, Tatsumi, Masayuki